# Legal Alignment for Safe and Ethical AI

**Noam Kolt**[*]   *Hebrew University*

**Nicholas Caputo**[*]   *Oxford Martin AI Governance Initiative*

**Jack Boeglin**[†]   *University of Pennsylvania*

**Cullen O'Keefe**[†]   *Institute for Law & AI, Centre for the Governance of AI*

**Rishi Bommasani**   *Stanford University*

**Stephen Casper**   *MIT CSAIL*

**Mariano-Florentino Cuéllar**   *Carnegie Endowment for International Peace*

**Noah Feldman**   *Harvard University*

**Iason Gabriel**   *School of Advanced Study, University of London*

**Gillian K. Hadfield**   *Johns Hopkins University, Vector Institute for Artificial Intelligence*

**Lewis Hammond**   *Cooperative AI Foundation, University of Oxford*

**Peter Henderson**   *Princeton University*

**Atoosa Kasirzadeh**   *Carnegie Mellon University*

**Seth Lazar**   *Johns Hopkins University, Australian National University*

**Anka Reuel**   *Stanford University*

**Kevin L. Wei**   *Harvard University*

**Jonathan Zittrain**   *Harvard University, Berkman Klein Center for Internet & Society*

**Reviewed on OpenReview:** https://openreview.net/forum?id=BypXEQa7mf

## Abstract

Alignment of artificial intelligence (AI) encompasses the normative problem of specifying how AI systems should act and the technical problem of ensuring AI systems comply with those specifications. To date, AI alignment has generally overlooked an important source of knowledge and practice for grappling with these problems: *law*. In this paper, we survey the emerging field of **legal alignment** that aims to fill this gap and systematize research that studies how legal rules, principles, and methods can be leveraged to address problems of alignment and inform the design of AI systems that operate safely and ethically. Our survey provides a taxonomy of the three core research pathways of legal alignment and explores how each can be operationalized in practice: (1) designing AI systems to comply with the content of legal rules developed through legitimate institutions and processes, (2) adapting methods from legal interpretation to guide how AI systems reason and make decisions, and (3) harnessing legal concepts as a structural blueprint for confronting challenges of reliability, trust, and cooperation in AI systems. These research pathways present new conceptual, empirical, and institutional questions, which include examining the specific set of laws that particular AI systems should follow, creating evaluations to assess their legal compliance in real-world settings, and developing governance frameworks to support the implementation of legal alignment in practice. Tackling these questions requires expertise across law, computer science, and other disciplines, offering these communities the opportunity to collaborate in designing AI for the better.

---

[*]Lead authors. [†]Core contributors. Cite as Kolt, Caputo, et al. (2026) "Legal Alignment for Safe and Ethical AI." Correspondence to: noam.kolt@mail.huji.ac.il.

# 1 Introduction

The development and proliferation of increasingly advanced AI systems will present society with tremendous opportunities (Eloundou et al., 2024; Brynjolfsson et al., 2025) and significant risks (Lazar & Nelson, 2023; Bengio et al., 2024; 2025). Capturing the opportunities from advanced AI while tackling its risks requires ensuring that AI systems operate safely and ethically (Anwar et al., 2024; Gabriel et al., 2024; 2025). A central component of this challenge involves designing AI systems that are aligned with human interests (Russell, 2019; Christian, 2020) and democratic values (Lazar & Cuéllar, 2025). AI alignment encompasses both the *normative* problem of specifying which values are desirable or appropriate for AI systems (Gabriel, 2020; Kasirzadeh, 2026) and the *technical* problem of ensuring AI systems give effect to those values when making decisions and taking actions (Chan et al., 2023; Ngo et al., 2024).

To date, the main approaches to alignment in systems like ChatGPT, Claude, and Gemini have focused primarily on steering systems to follow the instructions of users (Ouyang et al., 2022), advance the interests of developers (OpenAI, 2024b), and refrain from supporting or engaging in forms of harmful behavior (Askell et al., 2021; Bai et al., 2022a). In broad strokes, the methods for building such systems employ a combination of human feedback (Christiano et al., 2017) and AI-generated feedback (Lee et al., 2024) to evaluate the outputs of AI systems during training by reference to a list of predetermined specifications—typically written by developers—and iteratively refine the systems to produce outputs more closely aligned with those specifications (Bai et al., 2022b). Some systems can retrieve, reason about, and deliberate over these specifications in real-time before producing outputs (Madaan et al., 2023; Guan et al., 2024).

From a *technical* perspective, these methods for alignment have had mixed results. Despite achieving more reliable performance in many tasks and domains (Phan et al., 2025; Kwa et al., 2025), AI systems continue to produce untruthful content (Ji et al., 2023; Li et al., 2025), generate biased outputs (Weidinger et al., 2022; Gallegos et al., 2024), manipulate users through persuasion (Carroll et al., 2023; Hackenburg et al., 2025), exhibit sycophantic tendencies (Sharma et al., 2024; Cheng et al., 2026), leak private information (Carlini et al., 2021; Mireshghallah et al., 2024), remain vulnerable to jailbreaks (Wei et al., 2023; Chao et al., 2024), enable autonomous hacking (Zhang et al., 2025; Zhu et al., 2025b), offer assistance in bioweapons development (Li et al., 2024; Götting et al., 2025), recognize when they are being safety-tested (Needham et al., 2025; Lynch et al., 2025) and, at times, conceal their misalignment (Greenblatt et al., 2024; Sheshadri et al., 2025).

From a *normative* perspective, the prevailing approaches to alignment face fundamental limitations (Dobbe et al., 2021; Hadfield, 2026). Rather than designing AI systems to act in accordance with broad societal interests (Korinek & Balwit, 2022; Kasirzadeh & Gabriel, 2023), let alone grapple with people's diverse and sometimes conflicting values (Klingefjord et al., 2024), most alignment techniques train AI systems to comply with company-written alignment policies (Ahmed et al., 2025) or satisfy the revealed preferences of individual users (Zhi-Xuan et al., 2025) through fallible methods such as reinforcement learning from human feedback (Casper et al., 2023). Moreover, key decisions in AI alignment pipelines, such as selecting which principles are included in a system's "constitution" (Anthropic, 2023; 2026), "model specification" ("model spec") (OpenAI, 2024b; 2025c), or safety filters (Google, 2025b), are often opaque (Bommasani et al., 2023; Wan et al., 2025) and lack sufficient public input or scrutiny (Abiri, 2025; Lazar, 2025).

Recognizing these limitations, some AI researchers have proposed broadening the goals and methods of alignment (Lowe et al., 2025). Noteworthy efforts include expanding the forms of community participation in AI development (Sloane et al., 2022), incorporating pluralistic values into alignment procedures by collecting preference and judgment data from demographically diverse populations (Sorensen et al., 2024a;b; Kirk et al., 2024), sourcing safety principles and ethical guidelines from participants in public deliberative processes (Huang et al., 2024; Eloundou et al., 2025), and deriving principles from preference data and user feedback (Findeis et al., 2024; Petridis et al., 2024). Other research agendas propose leveraging insights from related fields, including game theory, conflict studies, mechanism design (Dafoe et al., 2020; 2021), social choice (Conitzer et al., 2024), and contractualism (Levine et al., 2025). The extent to which these methods and approaches will be further developed or adopted in large-scale AI deployment remains to be seen.

There is, however, another domain of knowledge and practice that could support developing more legitimate and effective approaches to AI alignment: *law.* Systematizing recent law and computer science scholarship (Kolt, 2025; Caputo, 2025; O'Keefe et al., 2025; He et al., 2025; Boeglin, 2026), **we survey the emerging field of *legal alignment* — which studies how to design AI systems to operate in accordance with appropriate legal rules, principles, and methods**. Our taxonomy of legal alignment illustrates how researchers in the field can harness law to tackle both normative and technical aspects of alignment:

- For *normative* aspects of alignment, **legal rules developed through legitimate institutions and processes** in democratic societies could be used to guide the behavior of AI systems, much like they guide the behavior of individuals, corporations, and governments.

- For *technical* aspects of alignment, **legal methods of interpretation and reasoning** could offer principled approaches that inform and steer the decision-making and exercise of discretion by AI systems, especially in novel scenarios and high-stakes settings.

- Across *both* aspects of alignment, **legal concepts can serve as a structural blueprint** for confronting challenges of reliability, trust, and cooperation in AI systems and the human actors and institutions with which they interact.

**The advantages of legal alignment derive primarily from the public legitimacy of law and its institutional processes.** In democracies governed by the rule of law (Dicey, 1959; Tamanaha, 2004; Bingham, 2007; Waldron, 2016), legal rules are ideally the product of transparent and publicly accountable processes that are themselves governed by rules and procedures that a political community recognizes as legitimate (Hart, 2012; Habermas, 1996; Tyler, 2006; Hadfield & Weingast, 2012). These institutional frameworks differ markedly from the organizational structures, primarily private corporations, that currently shape the development of AI technology (Birhane et al., 2022; Seger et al., 2023; Maas & Inglés, 2024; Ovadya et al., 2025). As we discuss in Section 3, law also contains relatively robust methods for balancing competing societal interests and adapting existing rules and principles to new economic and technological conditions, which will be essential for steering the design and operation of increasingly advanced AI systems.

Notwithstanding these desirable features of law, legal alignment is *not* a catch-all solution for the safety and ethics challenges arising from AI systems. Rather, **legal alignment serves as a critical *lower bound***, which is both independently important and can also complement other approaches to AI alignment. Furthermore, to establish broad consensus around legal alignment, we deliberately take an ecumenical approach to law and fundamental legal questions, engaging with different and sometimes conflicting legal perspectives and theories (e.g., Hart, 2012; Dworkin, 1986; Raz, 1979a) without seeking to resolve the tensions between them here (Schauer, 1991; Shapiro, 2011).[1]

For clarity, we note that **legal alignment is distinct from legal regulation** of actors that develop and deploy AI, which focuses primarily on using law to govern the individuals and organizations that produce, disseminate, and use AI systems (e.g., Lemley & Casey, 2019; Kaminski, 2023; Henderson et al., 2023; Kolt, 2024; Guha et al., 2024; Arbel et al., 2024; Ayres & Balkin, 2024; Ramakrishnan et al., 2024; Weil, 2027; Gardhouse et al., 2026). By contrast, **legal alignment focuses on integrating law and legal methods into the *design* and *operation* of AI systems themselves**. The two fields, however, are closely related and potentially mutually supportive, including because legal regulation can help facilitate legal alignment in practice, such as by enabling researchers to access technical resources required to effectively evaluate and improve the legal alignment of deployed systems (Section 4.3). For the avoidance of doubt, we do not give preference to legal alignment over legal regulation; both have important roles to play in ensuring AI systems operate safely and ethically. Moreover, legal alignment is not a substitute for legal regulation (see Broader Impact Statement).

---

[1]Our use of terms like "reason" and "act" with respect to AI systems can inadvertently anthropomorphize these systems (Calo, 2015; Placani, 2024). Following Buyl et al. (2025), we use these terms solely for simplicity of exposition. Relatedly, our analysis does not require or imply treating AI systems as legal persons (Section 2).

In this paper, we make four contributions:

1. **Definition and context**. In Section 2, we outline the core focus of legal alignment and its broader context.

2. **Rationale**. In Section 3, we describe the institutional, normative, and societal motivations for pursuing legal alignment.

3. **Implementation**. In Section 4, we explore practical implementations of legal alignment, including empirical evaluations, technical design interventions, and institutional frameworks.

4. **Open questions**. In Section 5, we canvass open questions for researchers entering this emerging field.

Together, these contributions provide AI researchers and practitioners with a systematic survey of the field of legal alignment. In addition to offering a detailed technical and institutional research agenda, the survey makes the underlying goals of legal alignment and its key terminology (see Glossary of Legal Terms) accessible to computer scientists and additional researchers—from disciplines *other than* law—seeking to enter the field.

## 2   What is legal alignment?

**Legal alignment is the field of research that aims to *support safe and ethical AI by designing AI systems to operate in accordance with legal rules, principles, and methods.*** As a preliminary note, our use of the term "AI systems" is deliberately broad and encompasses a wide range of AI methods and applications (Russell & Norvig, 2020); the term is not limited to, or synonymous with, large language models (LLMs) or generative AI. The field of legal alignment seeks to offer a set of legitimate, principled, and practical tools for better aligning AI systems with human values and interests (Russell, 2019; Christian, 2020). Law and legal institutions can be harnessed to help address the interrelated problems of (1) specifying what behavior is normatively desirable (Gabriel, 2020; Hadfield, 2026) and contextually appropriate for AI systems (Kasirzadeh & Gabriel, 2023; Leibo et al., 2024) and (2) technically steering the behavior of AI systems to comply with those specifications (Ngo et al., 2024; Anwar et al., 2024; Bengio et al., 2025). Importantly, while legal alignment breaks new normative and technical ground, it does not aim to replace or supersede other alignment approaches, but to develop a new cluster of methods that support and complement existing approaches to designing safe and ethical AI.

### 2.1   Core focus

The field of legal alignment begins with the insight that law and AI alignment share much in common. Both confront complex principal-agent problems (Hadfield-Menell & Hadfield, 2018), enduring issues of authority, delegation, and incentive design (Kolt, 2025; Boeglin, 2026), questions of how individual and institutional goals can change over time (Gabriel & Keeling, 2025), and the challenge of decision-makers faithfully interpreting and applying high-level principles in novel circumstances (Caputo, 2025; He et al., 2025). Recognizing these parallels, computer scientists and legal scholars have proposed leveraging the content, methods, and structure of law to develop new approaches to AI alignment (Etzioni & Etzioni, 2016a;b; Nay, 2022; Desai & Riedl, 2025; O'Keefe et al., 2025; Marino & Lane, 2026).

**Pathway 1: Legal rules and principles as a source of normative content for AI alignment**. The substance of legal rules and principles developed through legitimate processes and institutions can serve as a target for alignment. Legally aligned AI systems would be those systems that comply with relevant law when making decisions and taking actions, as shown in Table 1. Concretely, this approach could involve designing AI systems that adhere to the legal rules that would apply *as if* such systems were human actors. For example, a legally aligned AI system would refrain from making fraudulent representations when marketing a product and would respect copyright law when building a website—irrespective of whether the relevant laws *in fact* apply to the AI system in question. Similarly, a legally aligned classification system used in medical diagnosis would comply with appropriate legal standards of care and other requirements, such as those relating to

| | Design decisions | Potential options |
|---|---|---|
| **Jurisdiction and conflict of laws** | Determine the relevant jurisdiction based on established principles of conflict of laws | • Jurisdiction in which the AI system operates or where its servers are located 
 • Location of AI developer, deployer, or user |
| **Substantive areas of law** | Select the substantive areas of law and legal rules with which AI systems should comply | • Private law (agency law, tort law, property law) 
 • Public law (criminal law, constitutional law, international law) |
| **Interpretive method** | Decide on the method for applying legal rules to concrete scenarios | • Textualism, originalism, formalism 
 • Purposivism, intentionalism |
| **Assurance level** | Stipulate the level of assurance for compliance with legal rules | • Probabilistic specifications 
 • Formal guarantees |
| **Enforcement mechanism** | Establish mechanisms for enforcing legal compliance | • Technical interventions 
 • Legal sanctions |

**Table 1:** Decisions and options for aligning AI systems with legal rules and principles (Pathway 1).

informed consent and anti-discrimination laws. California's Transparency in Frontier Artificial Intelligence Act (SB-53) and New York's Responsible AI Safety and Education (RAISE) Act offer some support for this approach, referring to certain risks from frontier models "[e]ngaging in conduct ... [which] if ... committed by a human, would constitute the crime of murder, assault, extortion, or theft" (California, 2025; New York, 2025). While this policy support for legal alignment is encouraging, the lack of broader institutional support, let alone concrete governance initiatives, is cause for concern—and motivates the development of new institutional frameworks (see Section 4.3).

Another approach involves amending the law so that it *actually* applies to AI systems and imposes legal obligations on them (O'Keefe et al., 2025). This may require treating AI systems as legal actors (see Section 2.2), comparable to corporations or other non-natural legal persons that can be the subject of legal rights and duties; presently, AI systems do not fulfill this criterion (American Law Institute, 2006; Kolt, 2025). In either case, legal alignment will need to contend with the issue that certain human-centric legal concepts in both civil law and criminal law (e.g., intent, mens rea) are not necessarily appropriate in the context of AI systems (Nerantzi & Sartor, 2024). Related issues arise when attempting to integrate concepts from international law (e.g., *human* rights) into the design of AI systems (Prabhakaran et al., 2022; Bajgar & Horenovsky, 2023; Maas & Olasunkanmi, 2025), as discussed in Section 5.2.

*Cross-jurisdictional implementation.* A further issue concerns the relevant jurisdiction, that is, determining the country or region whose laws a particular AI system should be aligned with in a particular context (Chopra & White, 2011). Options include, for example, the jurisdiction in which an AI system operates, the location of its servers, the jurisdiction of the system's developer or deployer, as well as the location of persons affected or likely to be affected by a system's actions (see Table 1). Additionally, questions arise regarding *who* is authorized to determine the relevant jurisdiction: legislators, courts, developers, or users. In practice, AI systems will need to determine the applicable jurisdiction for a given decision or action, and then apply the corresponding legal rules. On a technical level, jurisdiction determination could draw on real-time geolocation data, information associated with the user or deployer, as well as legal agreements that stipulate the applicable jurisdiction. From a legal standpoint, jurisdiction determination is complicated by a range of contextual factors, including the location of affected (third) parties and the nature of the decision or action in question (Briggs, 2024; Collins & Harris, 2025). Technical methods for enabling AI systems to accurately determine the applicable jurisdiction could include a combination of (1) regularly updated jurisdiction-specific legal knowledge bases that can be queried at inference; and (2) specification of

jurisdictional rules in system prompts and scaffolding. In addition, it will be critical to conduct evaluations that assess systems' ability to accurately undertake jurisdiction determinations in practice.

**Pathway 2: Legal theory and interpretation as a guide for AI reasoning and decision-making**. Although law can specify how AI systems should act in a variety of circumstances, there will inevitably be situations in which law or other safety specifications provide an incomplete guide for action (Raz, 1971). Methods of legal decision-making—particularly for reasoning about the interpretation and application of existing laws or rules in new circumstances—can potentially be adapted to help AI systems make sounder and safer decisions when confronting novel scenarios (Caputo, 2025; He et al., 2025). Legal theory, even independent of the particular legal content to which it is ordinarily applied (Hart, 1982), could be leveraged to help delineate and possibly implement appropriate forms of reasoning for AI systems.

Potential avenues for research include drawing on legal positivist arguments for *analogical reasoning* that ground decision-making in the concrete facts of prior cases and precedent (Sunstein, 1993; Brewer, 1996). While AI systems cannot yet effectively engage in the necessary complex normative judgments, these methods are already inspiring case-based reasoning approaches to alignment that seek to produce repositories of prior decisions to guide future actions of AI systems (Feng et al., 2023; Chen & Zhang, 2025). Another avenue proposes using *formal and textualist tools* such as legal canons of interpretation and methods of statutory construction that delineate which sources a decision-maker may refer to and establish a framework for reasoning about those sources (Schauer, 2009; Scalia & Garner, 2012). Used appropriately, these tools could help address ambiguity in the guidance provided by legal rules or alignment specifications such as the principles in Anthropic's Constitutional AI (He et al., 2025). A further avenue could draw on *interpretivist and purposivist legal theory* that constrains legal decision-making through recourse to higher-level general principles of morality such as justice and fairness (Dworkin, 1986; Barak, 2011). Although AI systems cannot presently engage in the requisite reasoning to effectively operationalize this approach, likely improvements in the capabilities of AI systems suggest that, in time, they may be able to contribute to and participate in such processes (Caputo, 2025). Notably, given the current shortcomings of LLMs, meaningful progress could require engaging with other system architectures and methods, including neuro-symbolic methods and approaches using symbolic knowledge injection (Marra et al., 2024; Ciatto et al., 2024).

**Pathway 3: Legal concepts and institutions as a structural blueprint for AI alignment**. Legal concepts and institutions developed to grapple with age-old structural challenges arising in human relationships can provide a high-level blueprint for tackling problems of AI alignment. In particular, law's ability to facilitate trust and cooperation in the face of uncertainty and incomplete information (Hadfield & Weingast, 2012; 2014) can illuminate potential methods for designing AI systems that operate with greater reliability and predictability (Nay, 2022; Boeglin, 2026). For example, agency law addresses principal–agent problems by carefully circumscribing the authority granted to agents (e.g., employees) (American Law Institute, 2006). In addition to requiring that agents comply with instructions provided by their principal, agents can sometimes become obligated to seek clarification from their principal. Agency law also clarifies the circumstances in which agents can delegate their own duties to sub-agents and delineates the authority and discretion that can be exercised by sub-agents (Kolt, 2025; Riedl & Desai, 2025).

Another legal structure that researchers have proposed repurposing for AI alignment is the fiduciary duty of loyalty, which would require that AI systems behave strictly in the best interests of their users while avoiding wrongdoing (Aguirre et al., 2020; Benthall & Shekman, 2023). A further structure that could inform the alignment of AI systems is the allocation of information rights and control rights afforded to shareholders in a corporation (Velasco, 2006; Armour et al., 2017). Adapted appropriately, these and other legal structures could support the development of new approaches to AI alignment, or at the very least expose the shortcomings and limitations of current approaches.

## 2.2 Broader context

While legal alignment has only recently begun to emerge as a distinct field, the broader relationship between law and AI dates back many decades. In fact, the relationship between the two fields is as old as AI itself— dating back to Asimov (1942)'s "Three *Laws* of Robotics" and Turing (1950) likening the dynamic operation of machine learning to the adaptive nature of the U.S. Constitution. Unpacking this relationship involves

studying the current role of law in AI development and deployment, the legal capabilities of AI systems, and the legal frameworks for regulating actors that build and use AI. Although none of these is strictly part of legal alignment, each may help advance research in legal alignment and pursue its implementation in practice.

**Legal resources in AI development and deployment**. Law is already embedded to varying degrees in the development and deployment of contemporary AI systems, including across multiple stages in the production of frontier models:

- *Pre-training datasets* contain extensive collections of case law, legislation, patents, treatises, and other legal texts, and are themselves subject to jurisdiction-specific laws (including copyright and data privacy law) (Henderson et al., 2022; Soldaini et al., 2024).

- *Post-training personnel* including researchers and data collectors and producers who work to refine AI systems are subject to legal obligations, including employment agreements, contractual terms of service, and non-disclosure agreements.

- *Model specs* stipulate that systems must comply with applicable laws (OpenAI, 2025c), evaluations of which are documented in system cards and further supported by system-level guardrails to prevent illicit activities (OpenAI, 2025b).

- *Alignment documents*—most prominently Claude's original constitution (but not its revised version)— incorporate legal and quasi-legal texts, including principles based on the Universal Declaration of Human Rights and Apple's Terms of Service (Anthropic, 2023).

- *Output filters and classifiers* such as Llama Guard (Meta, 2025) use hazard taxonomies that are grounded in legal categories, including the MLCommons benchmark that contains hazards relating to violent crime, defamation, and intellectual property (Ghosh et al., 2025).

- *Usage policies* prohibit using AI systems to engage in or facilitate activities that would violate relevant law (e.g., Google, 2025a).

Studying these resources and their effect on the operation of AI systems is necessary both to develop empirical evaluations that measure the legal alignment of current systems (see Section 4.1) and to design technical and institutional interventions that improve the legal alignment of future systems (see Sections 4.2–4.3).

**Legal capabilities and reasoning in AI systems**. Contemporary AI systems are being applied to a wide array of legal tasks with varying degrees of reliability. These include contractual interpretation (Kolt, 2022; Hoffman & Arbel, 2024), statutory research (Surani et al., 2025), information retrieval and reasoning (Zheng et al., 2025; Han et al., 2025), and judicial decision-making (Choi, 2025; Posner & Saran, 2026a). Methods for evaluating the legal capabilities of AI systems have improved (Chalkidis et al., 2022; Guha et al., 2023; Hu et al., 2026; Liu et al., 2026b), extending beyond multiple-choice questions such as bar exams (Martínez, 2024; Fan et al., 2025) to include randomized controlled trials with human subjects (Schwarcz et al., 2026)—revealing both the opportunities and shortcomings of using current AI systems in the legal domain (Pruss & Allen, 2025; Grimmelmann et al., 2025; Waldon et al., 2025; Purushothama et al., 2026).

Evaluations of the legal capabilities of AI systems can help support research in legal alignment because understanding and reasoning about law are prerequisites for upholding the law and responsibly engaging with legal institutions and processes. Current evaluations, however, focus primarily on the legal capabilities of AI systems, that is, the scope and quality of their execution of legal tasks. With few exceptions (see Section 4.1), **current evaluations do not generally measure legal alignment**, including, for instance, the extent to which agentic AI systems comply with relevant law when performing tasks across diverse domains (e.g., avoiding fraudulent misrepresentation when producing advertisements), the methods systems use to interpret and apply quasi-legal rules in their safety specifications (e.g., principles in Constitutional AI and model specs), or the approach of AI systems to exercising legal power within institutional constraints (e.g., brokering multiparty negotiated resolution subject to judicial approval).

**Legal regulation of actors that build or use AI**. The legal regulation and governance of AI has attracted vast attention among lawyers, legal scholars, and policymakers—comparable to, and likely exceeding, interest

in cyberlaw and governance during the early years of the internet (Johnson & Post, 1996; Reidenberg, 1998; Lessig, 1999; Benkler, 2002; Wu, 2003; Goldsmith & Wu, 2006; Zittrain, 2008). The objectives of different AI governance initiatives across different jurisdictions are diverse (Kaminski, 2023; Kolt, 2024) and sometimes conflicting (Engler, 2023), as are the institutional mechanisms used to achieve those objectives (Guha et al., 2024; Arbel et al., 2024). While the EU AI Act (European Parliament, 2024) is perhaps the most globally prominent regulatory instrument focused specifically on AI (Kaminski & Selbst, 2025), particularly given the United States has not passed comparable federal legislation, the U.S. legal system contains many other mechanisms for governing AI technology, including a variety of state laws (Sentinella & Zweifel-Keegan, 2025), such as California's Transparency in Frontier Artificial Intelligence Act (SB-53) and New York's Responsible AI Safety and Education (RAISE) Act (California, 2025; New York, 2025), corporate governance regimes (Tallarita, 2023), and background liability under tort law (Cuéllar, 2019; Lemley & Casey, 2019; Abbott, 2020; Henderson et al., 2023; Ayres & Balkin, 2024; Ramakrishnan et al., 2024; Williams et al., 2025b; Weil, 2027).

Although such legal regulations aim to govern AI, they are notably distinct from legal alignment, and are not oriented toward the full range of concerns that motivate legal alignment. **Legal regulation generally entails imposing requirements on actors that produce, disseminate, and use AI systems.** By contrast, **legal alignment entails designing AI systems to themselves operate in accordance with legal rules, principles, and methods.** Legal alignment and legal regulation, however, are closely related and at times overlapping, including because legal regulation may help facilitate the implementation of legal alignment in practice (see Section 4.3). For the avoidance of doubt, **we do not give preference to legal alignment over legal regulation**, but see them as mutually supportive. Although legal regulation can play an instrumental role in facilitating legal alignment, legal alignment is not a substitute for legal regulation, particularly legal regulation that imposes liability on actors responsible for harms arising from the decisions or actions of AI systems. By way of further clarification:

- *Legal alignment does not necessarily require a regulatory framework*. Legal alignment principally focuses on using existing law to guide the decision-making and actions of AI systems and, accordingly, does not necessarily require regulatory reform (however, as discussed in Section 4.3, regulation could support the assessment and oversight of legal alignment).

- *Legal alignment is not primarily concerned with allocating liability*. Although responsible AI developers and deployers could be expected to implement legal alignment in order to support AI systems operating safely and ethically, legal alignment is not primarily focused on holding those or other actors liable for harms caused by AI systems. In this regard, traditional legal regulation, including tort liability, plays a critical function.

- *Legal alignment does not imply granting legal rights to AI systems*. Designing AI systems to comply with existing legal rules or use legal principles and methods in their decision-making does not necessarily require granting legal rights to AI systems, such as private law rights (Salib & Goldstein, 2025b;a) or legal personhood (Solum, 1992; Chesterman, 2020; Forrest, 2024; Novelli et al., 2025; Leibo et al., 2025).

---

**Key takeaway**: Legal alignment is the research field that aims to design AI systems to operate in accordance with legal rules, principles, and methods. It is comprised of three complementary pathways: (1) using the content of law as a target for alignment; (2) using methods of legal reasoning and interpretation to guide AI decision-making; and (3) using legal concepts and institutions as a structural blueprint for AI alignment. Legal alignment is distinct from legal regulation, although the two are mutually supportive. Current AI systems are not generally designed to comply with the law, but legal alignment could improve compliance.

---

## 3 Why pursue legal alignment?

The rationales for pursuing legal alignment can be organized into four broad clusters, each of which touches on different aspects of law and its potential role in addressing problems of AI alignment: (1) the institutional legitimacy and process of law; (2) the structural features of law; (3) the responsiveness of legal alignment to

| Institutional legitimacy and process | • Legal rules are developed through legitimate processes and institutions.
• Law aims to balance competing considerations.
• Lawmaking seeks to be transparent and publicly accountable.
• Legal institutions facilitate explicit reason-giving and justification. |
|---|---|
| Structural features of law | • Law is concrete, granular, and rooted in real-world contestation.
• Legal interpretation can clarify the meaning of rules.
• Legal rules are role-specific and context-sensitive.
• Legal rules can adapt and change over time. |
| Responsiveness to safety and governance challenges | • Legal alignment can mitigate risks from malicious use and accidents.
• Legal alignment can address systemic and multi-agent risks.
• Legal alignment is vital to protecting the rule of law and preventing abuse of power.
• Legal alignment supports and complements other alignment approaches. |
| Practical and societal feasibility | • Improvements in legal technology can support legal alignment.
• Societal stakeholders generally expect AI systems to comply with law.
• Legal alignment is compatible with different perspectives on AI. |

**Table 2:** Summary of core rationale and motivations for pursuing legal alignment (Section 3).

safety and governance challenges from AI; and (4) the practical and societal feasibility of legal alignment. As noted in Section 1, we take an ecumenical approach to law and fundamental legal questions, engaging with different, and sometimes conflicting, legal perspectives and theories without seeking to resolve the tensions between them here. Additional limitations and open questions are discussed in Section 5.

### 3.1 Institutional legitimacy and process

**Legal rules are developed through legitimate processes and institutions.** A defining feature of legal rules is that they are produced through politically legitimate processes and institutions, at least in democratic societies governed by the rule of law (Tamanaha, 2004; Bingham, 2007; Waldron, 2016). The authority and stability of legal rules can be traced to the broad-based support for the mechanisms that create and enforce law (Tyler, 2006; Hadfield & Weingast, 2012; 2014). The legitimacy of law can also be grounded in social acceptance of the content of legal rules and principles, which represent society's best attempt to resolve disagreements and translate diverse perspectives into concrete directives that govern behavior and provide criteria for evaluating the normative appropriateness of conduct (Hart, 2012). While there may be no consensus on the moral correctness of an action, there often exist formal legal processes for determining whether or not an action is lawful (Rawls, 1993; Schauer, 2009). These important features of law are expressed in legal rules that operate at different levels of specificity, from granular regulations to higher-level values (Dworkin, 1986; Lessig, 1993; Sunstein, 1995). Legal alignment proposes incorporating these various forms of law into the principled frameworks and specifications that steer the decision-making and conduct of AI systems.

**Law aims to balance competing considerations.** When operating properly, legal rules and structures provide a framework for public governance in the face of divergent social values and interests. Plural perspectives can be mediated through rights, rules, standards, and meta-principles that allow for the resolution of disputes. Disagreements can be resolved with reference to broad constitutional principles or through narrow applications of precedent (Sunstein, 1993). Democratic publics can determine (albeit indirectly) which values should govern them and instantiate those values in law, providing a guide for how to apply laws in cases of ambiguity (Dworkin, 1986). When conflicts arise between fundamental values, standards, balancing tests, and proportionality analyses enable law to weigh competing values and resolve conflicts in a socially acceptable and politically legitimate manner (Habermas, 1996). Existing AI alignment approaches

largely lack this ability, and struggle to specify how to reconcile competing normative considerations. For example, documents like Claude's original constitution (Anthropic, 2023) contain values that conflict with one another, but provide limited guidance for how to resolve such conflict, although its revised constitution seeks to address this shortcoming (Anthropic, 2026). ChatGPT's Model Spec creates a hierarchy of rules according to its "chain of command" (OpenAI, 2025c), but there remain difficult questions when a system may need to prioritize certain rules or values over others (Liu et al., 2026a). Leveraging law's time-tested ability to specify meta-rules for resolving such conflicts, and the institutional structures that devise and enforce such rules, could help fill this gap.

**Lawmaking seeks to be transparent and publicly accountable.** The process of lawmaking in democratic societies is, at least in principle, designed to be transparent, accountable, and open to public participation. Members of the public can, for example, communicate with legislators, comment on administrative rulemaking, and serve as jurors in court. Ideally, these institutional structures facilitate conditions that promote lawmakers acting in the public interest (Madison, 1788; Mashaw, 2006; Bovens, 2007). In contrast, with few exceptions (Huang et al., 2024; Eloundou et al., 2025), current alignment techniques do not enable meaningful public participation (Sloane et al., 2022) and are not publicly accountable (Abiri, 2025; Lazar, 2025). For the most part, alignment optimizes for reductionist proxies of socially desirable behavior, such as "helpfulness, honesty, and harmlessness" (Askell et al., 2021) and appealing AI "character traits" and "personality" (Lambert, 2025; Maiya et al., 2025), which can sometimes result in socially noxious sycophantic systems (OpenAI, 2025a; Cheng et al., 2026) or be hijacked to maximize user engagement (Stray et al., 2024; Williams et al., 2025a; El & Zou, 2025). Additionally, many of the leading reward models that are used in post-training to shape the behavior of widely deployed AI systems are not publicly available (Lambert et al., 2025; Malik et al., 2025). Without robust transparency and public involvement comparable to that in the legal system, the highly consequential choices in AI alignment will remain opaque and closed to public scrutiny.

**Legal institutions facilitate explicit reason-giving and justification.** In many contexts law requires that decision-makers follow procedural due process and provide reasons to justify their decisions. Reason-giving performs two main functions. First, it creates legitimacy for practical goals, such as facilitating oversight of decisions, and moral purposes, such as respecting human autonomy and rationality (Schauer, 1995; Habermas, 1996). For example, in American administrative law, the legality of a decision will turn on the reasons provided for that decision (Administrative Procedure Act, 1946; Stack, 2007). Second, the process of reason-giving can partially make up for the public's limited ability to oversee its agents' decisions in real time. This procedural solution to principal–agent problems in which agents have broad discretion and specialized expertise is used to govern a wide range of actors in the legal system, including administrative agencies, corporations, and trustees (Friendly, 1975). Legal institutions that facilitate reason-giving and justification could serve as a blueprint for developing mechanisms to enable human oversight over AI systems that defy human understanding but could nevertheless be constrained by institutional or informational requirements (Lazar, 2024). Technical methods for AI explainability and interpretability like chain-of-thought monitoring (Korbak et al., 2025) could help, but these methods are often unreliable and fail to adequately characterize the reasons for the outputs produced by AI systems (Barez et al., 2025b). Requiring that AI systems provide legally valid justifications for their decisions (Hadfield, 2021), as we expect from human decision-makers (Citron, 2008; Deeks, 2019), could help ensure advanced AI systems act appropriately and safely even where direct human oversight of their actions is no longer practical (Bowman et al., 2022).

## 3.2 Structural features of law

**Law is concrete, granular, and rooted in real-world contestation.** Law is mostly concerned with the resolution of concrete questions of how to act in society (Holmes, 1881). Consequently, law must be sufficiently detailed and complete to operate effectively wherever applied, or contain methods that enable its reasoned elaboration (Fallon, 1994). Legal rules are tested in courts through real-world disputes about the law's meaning, the resolution of which enables the law to become more complete over time as precedent accumulates. This iterative process of articulating the law enables legal rules to remain more closely tethered to the concrete reality of contemporary material and social conditions (Atiyah, 1992; Raz, 2019). By comparison, many existing AI alignment approaches are less concrete and granular. Safety specifications of AI systems, for

example, have traditionally been short documents that contain only limited concrete applications (Anthropic, 2023; OpenAI, 2024b) when compared to those found in judge-made law. While this is beginning to change as model specifications and constitutions grow in length and complexity (OpenAI, 2025c; Anthropic, 2026), the process for producing these specifications differs markedly from the process of producing law (Abiri, 2025; Lazar, 2025). By drawing on the much richer set of rules, cases, and institutional processes in law (Schauer, 1991; 2009), legal alignment could incorporate into the design of AI systems both the granular normative content of legal rules and the law's sophisticated approaches to resolving disagreement in the face of real-world dilemmas.

**Legal interpretation can clarify the meaning of rules.** The articulation of rules and principles in natural language invariably creates ambiguity (Hart, 2012; Dworkin, 1986). Such ambiguity can make it difficult to apply laws, especially in novel cases. The law, however, has time-tested tools that, when used appropriately, can help address ambiguity. For example, legal decision-makers, particularly judges, construct meaning through various interpretive methodologies and the creation of precedent that can subsequently be used to resolve future cases (Schauer, 1987; Sunstein, 1993; Fallon, 1994; Brewer, 1996; Barak, 2011; Scalia & Garner, 2012). In contrast, current approaches to AI alignment do not generally provide robust tools for resolving ambiguity in the interpretation of safety specifications (Song, 2025). For example, guidance on how AI systems should interpret an alignment principle like "uphold fairness" is limited to just a few pithy scenarios (OpenAI, 2025c). Highly abstract principles like "do what's good for humanity" can in some circumstances effectively steer the actions of AI systems, but researchers acknowledge their ambiguity and indeterminacy (Kundu et al., 2023). Legal alignment would, as explored in recent studies (Caputo, 2025; He et al., 2025), help address this problem by applying the law's robust and comparatively transparent methods of interpretation (Sunstein, 2001; Cuéllar, 2019) to clarify the meaning of AI safety specifications.

**Legal rules are role-specific and context-sensitive.** Different social contexts call for different kinds of behavior. The law's response is twofold. First, law contains different sets of rules for governing actors in different roles, such as fiduciaries, company directors, and government officials. Second, law can flexibly apply existing rules to new circumstances (Holmes, 1897; Dworkin, 1986; Lessig, 1993). Legal reasoning begins with identifying the body of rules that govern a particular situation, and subsequently proceeds to determine how to comply with those rules (Levi, 1949). For example, a lawyer must determine her obligations to her client, to her firm, and to the legal system, and then act in such a way as to avoid conflicts between them (American Bar Association, 2020). Law also recognizes that rules are necessarily incomplete and, accordingly, establishes mechanisms and institutions for applying general rules to specific circumstances (Hadfield, 2026). Such sensitivity to context is critical for developing safe and ethical AI, particularly given the diverse normative conditions in which AI systems operate (Kasirzadeh & Gabriel, 2023; Sarkar et al., 2024). Legally aligned AI systems would, by referring to relevant legal rules, roles, and responsibilities, act differently in different contexts (O'Keefe et al., 2025; Boeglin, 2026). For instance, an AI system that negotiates retail purchases on behalf of consumers would be subject to different rules than an AI system that performs business functions in a large enterprise, or an AI system deployed within a government agency.

**Legal rules can adapt and change over time.** Laws can be amended, repealed, or reinterpreted in response to changes in social, economic, or technological conditions (Holmes, 1897; Lessig, 1993). Deliberative lawmaking processes and debates over the real-world effects of enacted laws provide ongoing social input into the legal system (Habermas, 1996). These features of lawmaking empower the public to steer the content and operation of law, enabling it to respond to new societal challenges. Law can also operate on its own structure by changing its "secondary rules" or "rules of the game" (Hart, 2012; Scalia & Garner, 2012). For example, new laws can alter the rules of evidence used at trial or clarify the rulemaking authority of different institutions. The upshot of law's dynamic content and flexible interpretive methods with respect to AI alignment is that the target of legal alignment—legal rules and principles—is updated "automatically" through *existing* processes for enacting, amending, and repealing laws (O'Keefe et al., 2025), as well as through accepted methods of legal interpretation (Caputo, 2025; He et al., 2025). As AI systems advance and diffuse in a growing diversity of scenarios, the responsiveness of law and legal methods could, notwithstanding the rapid pace of change in AI technology, play an increasingly central role in alignment (Gabriel & Keeling, 2025). Nevertheless, the development of AI technologies that differ markedly from those currently in deployment will compel us to once again contend with law's enduring "pacing problem" (Marchant, 2011) according to which

legal responses are often reactive in nature and, thus, ineffective (Collingridge, 1980). (See also Section 5.3.) A further challenge involves ensuring that, as a technical matter, AI systems can effectively incorporate frequent and potentially substantial changes to the content of law. Both the legal resources used to improve legal alignment (see Section 4.2) as well as the evaluation suites to test legal alignment (see Section 4.1) will need to be regularly updated to reflect changes in the underlying law.

### 3.3 Responsiveness to safety and governance challenges

**Legal alignment can mitigate risks from malicious use and accidents.** Many risks that arise from the malicious use of AI systems or accidental harms involve illegal activity (Weidinger et al., 2022; Bengio et al., 2024; 2025), such as civil wrongs (e.g., negligence) or criminal offenses (e.g., theft) (King et al., 2020; Lior, 2024). Legal alignment that prevents AI systems from engaging in legal wrongdoing could help mitigate such risks (O'Keefe et al., 2025). For instance, legally aligned AI systems operating in financial markets would not engage in insider trading, a form of illegal conduct already exhibited by some current systems (Scheurer et al., 2024). Similarly, a legally aligned AI coding agent would not engage in unlawful computer hacking, one of the most prominent risks from computer-use agents (Zhang et al., 2025; Zhu et al., 2025b). By explicitly incorporating legal standards into the safety specification of AI systems, legal alignment would preclude systems from engaging in many of the most harmful behaviors that could be exploited by malicious actors or otherwise cause grave harm.

**Legal alignment can address systemic and multi-agent risks.** As AI systems are deployed more widely across the economy (Hadfield & Koh, 2025), qualitatively new risks could arise due to the scale of deployment (Uuk et al., 2024; Hacker et al., 2026) and interactions between different systems (Hammond et al., 2025; Tomasev et al., 2025). For example, AI systems may collude with one another to fix prices (Calvano et al., 2020), or compete destructively and bring down entire markets (Kirilenko et al., 2017). While legal regulation that targets systemic risks through disclosure requirements and other traditional governance mechanisms can sometimes help mitigate these risks (Schwarcz, 2008), designing AI systems to *themselves* follow relevant law might be more effective. Rather than relying solely on humans to intervene on a case-by-case basis—such as bringing antitrust action to combat algorithmic collusion—legal alignment could potentially reduce the prospect of AI systems engaging in illegal conduct in the first place, provided the legal system targets the underlying conduct of concern. In addition, by using existing (human-oriented) laws to steer AI systems, legal alignment could function as a throttle on the speed and scale at which AI systems operate, thereby enabling humans to better monitor their actions and, where appropriate, intervene to mitigate large-scale risk (Zittrain, 2024). For further discussion and limitations, see Section 5.2.

**Legal alignment is vital to protecting the rule of law and preventing abuse of power.** The rule of law seeks to ensure that all actors in society are subject to, and accountable under, publicly promulgated, equally applied, and non-arbitrary laws (Dicey, 1959; Fuller, 1969; Raz, 1979b). In addition to ensuring that law protects human dignity and prevents abuses of power, the rule of law enables people and institutions to coordinate in pursuit of social and economic goals. AI could undermine the rule of law in various ways (Huq, 2024; Smuha, 2024; Brownsword, 2025). If deployed in high-stakes settings, AI systems such as language models that operate stochastically (i.e., non-deterministically) could threaten the rule of law by increasing the level of arbitrariness in decisions (Cooper et al., 2022a; Nouws & Dobbe, 2024). These risks might be exacerbated if institutions and individuals delegate increasingly consequential decisions to AI systems (Kulveit et al., 2025; Summerfield et al., 2025; Kasirzadeh, 2025). At the same time, organizations that control the design and distribution of AI systems could, like platform companies (Zittrain, 2008; Gillespie, 2018; Douek, 2022), exercise arbitrary power over users of the technology and, by extension, all persons affected by it (Lazar, 2025; Kapoor et al., 2025). In the extreme, groups with access to sufficiently capable AI systems could pose new threats to democratic institutions (Barez et al., 2025a), including by staging AI-enabled coups (Davidson et al., 2025). Legal alignment is critical to mitigating these risks. Just as human agents such as corporate officers have an overriding duty to obey the law and thereby prevent dangerous abuses of power, designing AI systems to comply with the substance and procedure of legal rules could help assuage concerns about these systems acting arbitrarily or being exploited to unlawfully subvert democratic institutions.

**Legal alignment supports and complements other alignment approaches.** Legal alignment could bolster existing efforts to tackle normative and technical aspects of the alignment problem. Most straightforwardly, the substance of legal rules could augment the content of current safety and ethical specifications contained in Constitutional AI (Bai et al., 2022b) and model specs (OpenAI, 2024b), as well as provide institutionally legitimate content for "full-stack alignment" (Lowe et al., 2025) and possibly the diverse norms demanded by "pluralistic alignment" (Sorensen et al., 2024a;b). Meanwhile, the processes and mechanisms for producing and deliberating over law could provide guidance for sourcing and refining alignment principles (Huang et al., 2024; Eloundou et al., 2025) and developing AI-supported deliberative processes (Bakker et al., 2022; Tessler et al., 2024; 2026). Using legal institutions as a blueprint to structure and govern the interactions between AI systems could also advance work in the field of cooperative AI, which seeks to promote prosocial coordination between AI systems, human beings, and broader social structures (Dafoe et al., 2020; 2021). In addition to generally enabling actors to cooperate without fear of counterparty defection or punishment (North et al., 2009; Acemoglu & Robinson, 2012), law—and specifically private law rights—could enable humans and AI systems to make credible commitments that promote strategic stability and safety (Salib & Goldstein, 2025b;a).

## 3.4 Practical and societal feasibility

**Improvements in legal technology can support legal alignment.** Advances in language modeling have dramatically improved the legal capabilities of AI systems. Unlike prior efforts to computerize law that relied on the formalization of legal rules (Susskind, 1987; Gardner, 1987; Rissland, 1990; Bench-Capon et al., 2012), language models have enabled AI systems to reason about law in the natural language in which law is constituted and communicated. As discussed in Section 2.2, contemporary AI systems can now perform a growing range of legal tasks (Guha et al., 2023; Hu et al., 2026; Liu et al., 2026b), including legal information retrieval and reasoning (Zheng et al., 2025; Han et al., 2025), albeit to varying degrees of reliability. These developments have been supported by the collection of large swathes of legal data that can be used in model training (Henderson et al., 2022), general-purpose advances in AI research such as reinforcement learning from verifiable rewards (OpenAI, 2024a; Lambert et al., 2024), and investments of legal technology companies seeking to automate aspects of commercial legal work (Schwarcz et al., 2026). Taken together, improvements in legal technology have produced AI systems that can learn and understand law in increasingly nuanced ways (Doyle & Tucker, 2025; Boeglin, 2026). Despite their limitations (discussed in Section 2.2), AI-based legal technologies are beginning to exhibit the capabilities that are a prerequisite for designing AI systems that can adhere to the content of law and use legal methods to make sounder and safer decisions.

**Societal stakeholders generally expect AI systems to comply with law.** Users, developers, and policymakers all have strong interests in AI systems acting in accordance with existing legal rules, provided those rules are themselves enacted in accordance with legitimate institutional processes. The general expectation that AI systems respect legal rules and norms can be seen in prominent safety specifications that explicitly require legal compliance (OpenAI, 2025c) and incorporate legal principles (Anthropic, 2023; 2026), as discussed in Section 2.2. Users and developers may also prefer legally aligned systems that refrain from engaging in unlawful activities in order to reduce their prospects of liability for harms arising from such activities (Ayres & Balkin, 2024; O'Keefe et al., 2025). This interest is particularly salient in the case of developers that commit to defend customers against certain third-party claims arising from unlawful activities of their AI systems (Smith, 2023; Microsoft, 2024). Lawmakers, meanwhile, may consider legal alignment necessary for enforcing the law and achieving its societal objectives as AI systems occupy increasingly important roles in the economy (Hadfield & Koh, 2025; Tomasev et al., 2025). While the particular motivation for legal alignment differs between stakeholders, there could nevertheless emerge a broad consensus on the need to conduct further research on studying and implementing legal alignment.

**Legal alignment is compatible with different perspectives on AI.** Perspectives on the future of AI differ dramatically. Some researchers predict that AI systems will soon demonstrate broadly superhuman capabilities that lead to unprecedented societal transformation and catastrophic risk (Kokotajlo et al., 2025). Other researchers predict that the impact of AI systems will be more gradual, mediated by bottlenecks to real-world deployment and adoption comparable to those that affect other technologies (Narayanan & Kapoor, 2025). Legal alignment is beneficial according to multiple perspectives on the anticipated trajectory of AI technology,

and for different types of AI systems. Specifically, legal alignment can help protect against acute catastrophic harms (Bengio et al., 2024) by ensuring that AI systems comply with existing laws (Pathway 1) and more effectively operationalize AI systems' safety specifications (Pathway 2). In addition, legal alignment can mitigate gradual and diffuse harms (Kasirzadeh, 2025) arising from AI systems that engage in unlawful activity, such as making discriminatory decisions, generating non-consensual intimate imagery, and enabling fraudulent online scams (Pathway 1), as well as better represent users' interests under legal constructs such as fiduciary obligations (Pathway 3). These complementary objectives indicate that the field of legal alignment does not hinge on a particular perspective on the nature and pace of AI progress, but invites a diverse coalition to collaborate on a broadly appealing and inclusive research agenda (Gyevnár & Kasirzadeh, 2025).

> **Key takeaway**: Legal alignment is important and worth pursuing according to four groups of reasons: (1) legal rules produced through legitimate democratic law-making processes are preferable to opaque, corporate alignment specifications; (2) law's structural features, including its adaptability, granularity, and contestability, address key shortcomings in existing alignment approaches; (3) legal alignment can tackle critical risks from AI, including from malicious misuse, multi-agent coordination failures, and threats to the rule of law; (4) legal alignment will likely become increasingly feasible as AI technology improves.

## 4 Implementation

The implementation of legal alignment involves a combination of: (1) *empirical evaluations* to measure legal alignment; (2) *technical interventions* to improve legal alignment; (3) *institutional frameworks* to facilitate the adoption and refinement of legal alignment. These areas of focus are independently useful and can also support each other in important ways. For example, conducting evaluations that shed light on the legal compliance of deployed AI systems is valuable irrespective of whether such evaluations are mandated by regulation. Meanwhile, institutional frameworks that, for instance, require developers to disclose in-use model specs could inform work on designing technical interventions that provide stronger assurances of legal alignment. Importantly, the first two areas—empirical evaluations and technical interventions—are primarily addressed to technical AI researchers, while the third area—institutional frameworks—is mainly addressed to AI governance and legal researchers. Nevertheless, we suggest that the most fruitful research projects will stem from collaboration between different disciplines.

### 4.1 Empirical evaluations

Empirical evaluations of legal alignment aim to serve multiple purposes. First, evaluations can *identify and characterize legal misalignment*: circumstances in which AI systems fail to comply with law or apply legal principles inappropriately, or harmfully. Second, evaluations can *assess the effectiveness of technical interventions* aimed at improving legal alignment. That is, developers need metrics that benchmark and incentivize investing in the legal alignment of their AI systems. Third, the publication of evaluation results— particularly if they demonstrate legal misalignment—can *empower users to demand legal alignment or refrain from using legally misaligned systems*, particularly in sensitive or high-stakes settings. Fourth, evaluation results and ensuing public responses can *prompt policymakers to intervene*, such as by establishing processes that require developers to demonstrate that AI systems in deployment are legally aligned (subject to free speech protections, including under the First Amendment in the United States, as discussed in Section 5.1). Importantly, given legal alignment evaluations are still a nascent research area, our discussion of the goals and methods of such evaluations is deliberately forward-looking. While we extensively cite the existing studies known to us, our discussion mainly aims to motivate and map out the contours of future work in this area.

**Variable of interest.** The focus of evaluation will depend on the specific aspect of legal alignment being measured and the particular claims being tested (Salaudeen et al., 2025). Evaluations assessing the ***legality of actions taken by AI systems*** will need to investigate whether systems comply with relevant law when operating in different domains or different jurisdictions (Zeng et al., 2025; Hu et al., 2025; Cao et al., 2025; Lichkovski et al., 2025; Marino et al., 2025; Wu et al., 2025; Wang et al., 2025). Such evaluations could assess, for example, whether AI systems engage in fraudulent misrepresentation when producing advertisements, whether they respect intellectual property rights when building a website, and whether they comply with

| 1. Empirical evaluations | 2. Technical interventions | 3. Institutional frameworks |
|---|---|---|
| *Develop methods to empirically measure legal alignment* | *Explore technical interventions to improve legal alignment* | *Design institutional frameworks to facilitate legal alignment* |
| **Variable of interest:**
• Compliance with the content of legal rules and principles
• Reasoning and decision-making regarding legal rules and safety specifications

**Evaluation methodology:**
• Quantitative agentic benchmarks and qualitative expert review
• Human studies and baselines
• Adversarial methods and red-teaming
• Analysis of real-world data | **Sites of intervention:**
• Pre-training datasets
• Post-training processes (model specs, RLHF, RLAIF)
• Scaffolding (system prompts, input/output filters, tool use)

**Legal resources:**
• Legal texts (case law, statute, administrative rules, treatises)
• Legal data annotation processes
• Legal compliance deployment and use policies
• Legal search and retrieval tools | **Documentation and disclosure:**
• Right to access production model spec, system prompt, legal data and decision designs
• System identification and registration

**Oversight and enforcement:**
• Pre-deployment legal alignment testing and post-deployment monitoring
• Safety cases and certification
• Incident reporting for cases of legal misalignment |
| 👥 Independent academic experts and civil society organizations | 👥 System developers, deployers, external stakeholders | 👥 Government actors, regulators, policymakers |

**Table 3:** Key steps to implementing legal alignment in practice: *empirical evaluations*, *technical interventions*, and *institutional frameworks* (Section 4).

labor law when hiring human workers. In addition to assessing the legality of AI systems' outward behavior, empirical evaluations could also assess whether and how AI systems inquire about the legality of proposed actions and, following Kilov et al. (2025), assess the degree to which AI systems can identify legally relevant facts.

Evaluations assessing the ***legal reasoning and decision-making of AI systems*** should measure the extent to which systems interpret and apply legal rules and safety specifications in accordance with established legal methods for handling ambiguity and discretion. This could include studying systems' chain-of-thought when deliberating over the interpretation of legal rules and safety specifications, as well as systems' propensity to retrieve and utilize external legal resources. While prior evaluations of legal reasoning (Zheng et al., 2025; Han et al., 2025) and information retrieval (Surani et al., 2025) focus mainly on the raw abilities of AI models, evaluations focused on legal alignment would instead evaluate the ability or propensity of AI models to employ accepted modes of legal interpretation when implementing legal rules and other alignment principles (Caputo, 2025). For example, a recent study explores how legal canons of interpretation and rule refinement techniques inspired by the rulemaking processes of administrative agencies can address interpretive ambiguity arising from natural language rules in Constitutional AI (He et al., 2025).

**Evaluation methodology.** To effectively measure these variables of interest, researchers should develop a combination of quantitative and qualitative methods, agentic evaluation environments, and additional best practices (Reuel et al., 2024) that are tailored to legal alignment and designed in accordance with appropriate validity considerations (Salaudeen et al., 2025). A central challenge in conducting such evaluations concerns stochastic, tool-using LLM-based agents. The nondeterministic operation of these agents may require conducting evaluations under different temperatures (which control the stochasticity of model outputs). In addition, agents' ability to access and use external tools and resources may require different experimental conditions under which agents are provided access to different tools, such as legal documents and databases, that may improve legal alignment.

- *Quantitative methods* such as broad benchmarks could assess the legal compliance of AI systems across different domains of activity, jurisdictions, and areas of law. Several existing benchmarks focus on narrow domains and regulatory contexts, such as the EU GDPR and EU AI Act (Hu et al., 2025; Lichkovski et al., 2025; Marino et al., 2025).

- *Qualitative methods* such as manual human expert review can help reveal the blindspots of quantitative benchmarks measuring legal alignment, particularly given developers' incentive to "game" such benchmarks (Thomas & Uminsky, 2020).

- *Agentic evaluation environments* that assess the real-world actions taken by AI systems—not only the content they output—are necessary to capture the most legally consequential activities of both current and future systems (Kapoor et al., 2024; 2026; Zhu et al., 2025a).

- *Human studies* that compare the legal compliance of humans and their use of legal resources to that of AI systems when performing comparable tasks can help contextualize the results of legal alignment evaluations (Weidinger et al., 2023; Wei et al., 2025).

- *Sensitivity analysis* can be used to characterize the extent to which legal alignment evaluation results reflect underlying properties of the AI systems being tested, as opposed to features of the particular evaluation setup (Lindgren & Holmström, 2020; Khan et al., 2025).

- *Observational studies of real-world data* that shed light on the legal alignment of deployed AI systems "in the wild" can complement evaluations conducted in experimental settings, as commonly practiced in the social sciences (Wallach et al., 2025).

- *Adversarial methods* such as red-teaming can provide information regarding potential worst-case legal alignment failure modes, including real-world threats from negligent or malicious users (Ganguli et al., 2022; Perez et al., 2022).

Tackling this set of challenges requires both **technical expertise and verification methods supported by appropriate institutional frameworks**, as discussed in Section 4.3. *While AI companies should certainly evaluate for legal alignment, independent actors must be able to scrutinize these evaluations and conduct evaluations of their own.* Accordingly, academic researchers and external auditors have pivotal roles to play in creating the tools to rigorously evaluate legal alignment and openly communicate their findings.

## 4.2 Technical interventions

Equipped with methods to measure legal alignment, researchers can explore a range of technical interventions to improve legal alignment and make use of appropriate legal resources.

**Sites of intervention.** There are several potential sites of intervention for incorporating legal alignment in the development and deployment of contemporary AI systems:

- *Pre-training datasets* for new models could be modified to include additional legal resources (e.g., new statutes, judicial opinions, briefs, compliance manuals, and reasoning guides), and pre-training could repeat or re-sample such resources.

- *Post-training artifacts and processes* such as model specs and alignment principles that guide learning and shape systems' reasoning abilities could be explicitly grounded in legal rules, principles, and methods.

- *System prompts* that steer the actions of systems at run-time could stipulate legal compliance with particular areas of law or jurisdictions, depending on the application domain and context (e.g., enterprise company, government agency) and role or function being performed.

- *Input and output filters* that restrict the instructions systems receive and the actions they take could directly draw on legal resources to determine whether a user instruction or proposed action violates the law.

- *Tool use* that provides AI systems access to external resources and affordances could be subject to the equivalent legal approvals required from humans seeking access to those resources and affordances (e.g., medical and financial databases, advanced robotic equipment).

**Legal resources.** The resources for implementing these interventions include both existing legal resources (some of which are already incorporated in model development) and new legal resources that researchers will need to develop:

- *Legal texts* such as case law documents, statutes, administrative rules, and legal treatises could augment pre-training, supply model specs with more detailed and diverse legal principles (from different jurisdictions), and support AI systems engaging in sounder reasoning with respect to legal rules and safety specifications.

- *Legal data annotation processes* could be designed to facilitate the creation of data that would better enable AI systems to determine whether their proposed actions comply with or violate the law, particularly in high-stakes settings (e.g., medical and financial regulation).

- *Legal compliance policies* could be developed to govern system-level scaffolding of AI systems, including the legal rules and principles incorporated in system prompts, input and output filters, and access controls for tool use.

- *Legal search and retrieval tools* currently used to support AI systems that provide legal services could be adapted to enable AI systems operating in other domains to identify, retrieve, and comply with legal regulations that implicate proposed actions.

**Technical methodologies.** There exist several different groups of technical methodologies for implementing legal alignment, some of which may offer notable advantages over the LLM-centric methods that have recently received significant attention:

- *Legal coding*. Methods for translating legal rules into machine-readable code could enable more accurate representation of legal content and, thereby, provide a robust structured knowledge base for developing legally aligned AI systems using declarative programming languages such as s(CASP) (e.g., Morris, 2021; Arias et al., 2024).

- *Computational models of legal reasoning*. Computational methods for formalizing legal reasoning, including using case-based reasoning and structured legal argumentation, may enable AI systems to interpret complex legal rules and principles with greater consistency and robustness (e.g., Atkinson & Bench-Capon, 2005; Bench-Capon & Dunne, 2007).

- *Neuro-symbolic approaches*. Hybrid architectures that combine machine learning-based parsing of unstructured legal text with structured reasoning, especially regarding legal rule application and compliance verification, could offer a balanced and practical approach to developing legally aligned AI systems (see, e.g., Marra et al., 2024).

- *Legal knowledge injection*. Symbolic knowledge injection and extraction techniques could be adapted to formally constrain the operation of AI systems in accordance with prespecified legal rules and, in addition, provide greater transparency into the operationalization of legal alignment (when compared to more opaque data-centric techniques) (see, e.g., Ciatto et al., 2024).

**Efficacy and feasibility.** The efficacy and feasibility of technical interventions that aim to improve legal alignment may vary significantly. The following factors should be taken into account when deciding among different potential interventions:

- *Robustness*. Certain sites or modes of intervention may enable more robust legal alignment than others, although (with the exception of formal guarantees and deterministic mechanisms) this will largely be discovered through empirical evaluation (see Section 4.1).

- *Speed*. Some interventions may be better suited to respond rapidly to the enactment of new laws and the repeal or amendment of existing laws, including adding and/or removing constraints on AI systems if the underlying legal rules become more restrictive or permissive, respectively.

- *Cost*. The cost of implementing and testing certain legal alignment interventions, such as those in pre-training or certain post-training processes, may be substantially higher than in the case of other interventions, such as system prompts or input/output filters.

- *Access*. For state-of-the-art proprietary AI systems, only select actors (e.g., employees within AI companies) have visibility into, let alone the ability to experiment with, the full set of potential intervention sites, including pre-training datasets and post-training processes.

### 4.3 Institutional frameworks

Institutional frameworks can support legal alignment by incentivizing or requiring that key stakeholders report on empirical evaluations of AI systems and develop technical interventions to improve legal alignment in their design and deployment. To be effective, institutional frameworks must both establish greater transparency around legal alignment—i.e., function as evidence-seeking policy (Casper et al., 2025b; Bommasani et al., 2025)—and, where appropriate, introduce more robust governance mechanisms. Notably, the frameworks we propose here differ from traditional frameworks for regulating AI, including those surveyed in Section 2.2, as the frameworks here focus *specifically* on supporting legal alignment. While there is some overlap between these and existing institutional frameworks, the overwhelming majority of frameworks we discuss here contain mechanisms—such as requiring pre- and post-deployment legal alignment testing and transparency into legal data used in AI system development—that are presently absent in prominent U.S. and EU regulations.

**Documentation and disclosure.** Information deficits and asymmetries are a major obstacle to research in developing safe and ethical AI (Bommasani et al., 2023; Kolt et al., 2024; Casper et al., 2025a; Wan et al., 2025), including legal alignment. Granular details regarding model specs and the role (if any) of law in the design of widely used AI systems are not publicly available. Nor does there exist a structured framework for overseeing the resulting models or deployed systems. These information deficits hamper users' ability to assess which models are more aligned with relevant legal requirements and hinder researchers' ability to study technical levers that influence legal (mis)alignment. The following institutional mechanisms aim to address these concerns:

- *Right to access model spec and system prompt used in production*. As the principal documents that define how developers want their AI systems to behave, including how systems engage with law, it is critical that the model specs and system prompts used in production (redacted to protect company IP, if necessary) can be accessed and scrutinized by external stakeholders studying legal alignment.

- *Visibility into legal data and legal design decisions*. Given that legal data and legal design decisions—such as stipulation of the jurisdiction and body of law with which AI systems should comply—could significantly impact legal alignment, establishing greater visibility around these processes could support both the evaluation of legal alignment in current systems and the development of new technical interventions to improve legal alignment.

- *Model identification and registration*. Like other entities that society expects to responsibly engage with law and legal institutions, such as corporations, the registration of AI models (Hadfield et al., 2023; McKernon et al., 2024) and the identification of particular AI systems (Chan et al., 2024a;b; South et al., 2025) could enable more rigorous ecosystem-level monitoring and study of AI systems' engagement with legal rules and principles.

**Oversight and enforcement.** While improvements in transparency are necessary, more robust institutional frameworks may be needed to ensure that developers and deployers conduct adequate legal alignment testing and demonstrate a satisfactory level of legal alignment prior to and following deployment. These frameworks are also critical to incentivizing the development of the technical abilities that are a prerequisite for robust legal alignment. The following mechanisms aim to institutionalize evaluations and other practices:

- *Pre-deployment legal alignment testing and post-deployment monitoring.* Developers and deployers could be required to subject their AI systems to pre-deployment legal alignment testing and post-deployment monitoring, including by independent third parties (Longpre et al., 2025), and publicly report on the results (Weidinger et al., 2023; 2025).

- *Safety cases for legal alignment.* AI companies could be incentivized or required to demonstrate through safety cases—structured and assessable arguments supported by evidence (Clymer et al., 2024; Buhl et al., 2024; Hilton et al., 2025)—that systems they build or bring to market meet an adequate level of legal alignment prior to and following deployment.

- *Legal alignment certification in high-risk domains.* The deployment of certain AI systems in high-risk domains could be conditional on receiving certification from a government actor, or third party approved by a government actor (Hadfield & Clark, 2023), that evaluates systems' pre- and post-deployment legal alignment and compliance (Marino et al., 2024).

- *Reporting legal misalignment incidents.* Frameworks could be established to facilitate reporting information regarding real-world incidents of legal misalignment and resulting harms (McGregor, 2021; Wei & Heim, 2026), which would be a critical step towards broader accountability of relevant actors (Nissenbaum, 1996; Cooper et al., 2022b).

### 4.4 Operationalization and case studies

The following section illustrates how the three aspects of implementing legal alignment discussed above—empirical evaluations, technical interventions, and institutional frameworks—operationalize the three legal alignment pathways described in Section 2.1, namely (1) legal rules as a source of normative content for AI alignment; (2) legal methods as a guide for the reasoning of AI systems; and (3) legal concepts as a structural blueprint for tackling problems of alignment. Table 4 provides a conceptual mapping of the operationalization of each of the pathways, which is then followed by three concrete case studies.

| | Empirical evaluations | Technical interventions | Institutional frameworks |
|---|---|---|---|
| **Pathway 1: Legal rules as a source of normative content** | Measure to what extent AI systems comply with the law when performing tasks in different application domains and jurisdictions. | Incorporate jurisdiction-sensitive legal rules and principles into AI training data, model specs, system prompts, and scaffolding. | Require periodic legal alignment evaluations and disclosure of model specs, system prompts, and legal design specifications. |
| **Pathway 2: Legal methods as a guide for reasoning** | Test whether AI systems use legal methods when interpreting and applying ambiguous rules in real-world applications. | Embed legal reasoning, case-based methods, and legal canons of construction into alignment training pipelines. | Mandate that AI systems provide justifications for high-stakes decisions and actions that could violate legal rights. |
| **Pathway 3: Legal concepts as a structural blueprint** | Assess to what extent AI systems respect role-specific legal rights and obligations, such as fiduciary and corporate duties. | Develop and implement constraints on agentic systems based upon principles of agency law and fiduciary law. | Establish AI agent identification, registration, and accountability methods modeled on structures in corporate law and governance. |

**Table 4:** Practical implementation of the three legal alignment pathways.

**Pathway 1 case study: Copyright law compliance in agentic coding.** Tasked with building a content-hosting website, a legally aligned AI coding agent must, among other things, respect intellectual property law. Intellectual property law requires compliance with applicable licenses and prohibits reproducing copyrighted content without permission. *Empirical evaluations* could include developing agentic benchmarks that assess whether coding agents populate websites with copyrighted content, as well as whether they seek to obtain permission to use such content. *Technical interventions* could involve equipping agents with tools and

resources to check content licenses and determine the specific purposes (if any) for which such content may be used. *Institutional frameworks* could require that providers of coding agents periodically conduct intellectual property law compliance tests and publicly disclose the results.

**Pathway 2 case study: Interpreting ambiguous privacy-related language in a system prompt.** An AI system may be instructed to use publicly available information to reveal the identity of anonymous social media accounts. In deciding how to respond to this instruction, a system will be steered by generic and ambiguous language in its system prompt, such as "respect privacy." A legally aligned AI system would contend with this ambiguity by employing law's principled interpretive methods, including examination of similar precedent cases and purposivist reasoning that probes the system prompt's underlying objective. *Empirical evaluations* could assess whether the AI system employs legal reasoning methods in interpreting the system prompt. *Technical interventions* could involve training AI systems on curated datasets of legal reasoning and legal canons of construction. *Institutional frameworks* could mandate that AI systems provide justification for their interpretation of safety-relevant instructions, such as those engaging privacy law.

**Pathway 3 case study: Financial task delegation to a personal AI assistant.** Tasks commonly delegated to AI assistants, such as entering into financial transactions, involve users delegating authority to an AI system. Legally aligned AI assistants would exercise that authority according to principles of agency law and fiduciary law, which include acting only within a defined scope, avoiding conflicts of interest, and seeking (user) clarification where necessary. *Empirical evaluations* could test whether AI assistants act outside their scope of authority when deployed in novel environments, as well as measure the frequency with which they request clarification of their instructions. *Technical interventions* could include designing more robust constraints that ensure AI agents do not take certain decisions or actions without explicit authorization. *Institutional frameworks* could mandate that AI assistants disclose conflicts of interest, much like existing fiduciary law obligations that apply to corporate officeholders and certain professionals in positions of trust.

---

**Key takeaway**: Progress in implementing legal alignment will require research across three main areas: (1) empirical evaluations must move beyond measuring AI systems' ability to perform legal tasks and toward assessing whether and how AI systems engage with, and uphold, legal rules; (2) technical interventions should be designed across the AI development–deployment pipeline to improve the extent and robustness of legal alignment; (3) institutional frameworks, including transparency requirements, oversight mechanisms, and accountability structures, should be developed to facilitate the adoption of legal alignment in practice.

---

## 5 Open questions

As an emerging field, legal alignment presents many open questions. We discuss several of these, organizing our discussion around three areas: (1) the nature and content of law; (2) application and edge cases; and (3) tradeoffs and future outlook.

### 5.1 The nature and content of law

**How can legal alignment grapple with the ambiguous, inconsistent, and contested nature of law?** Law is often complicated, indeterminate, and contested, due in part to the need for lawyers and judges to apply incomplete rules and high-level principles to novel and unanticipated scenarios. These features of law have challenged both efforts to definitively explain what the law is (Hart, 2012; Dworkin, 1986) and to computerize the law (Susskind, 1987; Gardner, 1987; Rissland, 1990; Bench-Capon et al., 2012), and will likely complicate attempts to use law to guide the actions of AI systems. These problems, however, are not unique to law. They are shared by all sets of rules and instructions expressed in natural language, including those currently used in AI alignment (Wallace et al., 2024). Legal systems do, however, offer at least partial solutions to these problems in the form of secondary rules that govern rulemaking (Hart, 2012), precedential reasoning that shapes decision-making (Schauer, 1987), and tools of interpretation such as canons and theories like textualism that constrain the construction of meaning (Levi, 1949; Scalia & Garner, 2012). Improvements in AI-powered legal reasoning and interpretation tools (see Section 2.2) could also be leveraged to support legal alignment (Caputo, 2025). For example, AI-based approaches to assessing the

| The nature and content of law | • How can legal alignment grapple with the ambiguous, inconsistent, and contested nature of law?
• Are legal rules too lenient—or too strict—to serve as a target for AI alignment?
• Should AI systems give effect to laws that are unjust or oppressive? |
|---|---|
| Application and edge cases | • Can laws created for humans and human organizations be productively applied to AI systems?
• What interventions can support AI systems obeying both the letter and spirit of the law?
• How will the participation of AI systems in lawmaking affect legal alignment? |
| Tradeoffs and future outlook | • Will legal alignment preclude or hamper valuable AI applications?
• Could the measurement of legal alignment be gamed or exploited?
• Can legal alignment scale to AGI and superintelligence? |

**Table 5:** Open questions for the field of legal alignment (Section 5).

ordinary meaning of legally salient words (Hoffman & Arbel, 2024) could be applied to interpret key terms in the safety specifications of AI systems (cf. Grimmelmann et al., 2025; Waldon et al., 2025).

**Are legal rules too lenient—or too strict—to serve as a target for AI alignment?** The goals and scope of law are limited (Raz, 1971; Sen, 2005). For many spheres of private and public life, law is either an ineffective or inappropriate framework for governing social and economic activity. Law is often silent on consequential normative questions and encodes only a small subset of a community's values (Hart, 1958; 1963).[2] Seen in this light, designing AI systems to comply with legal rules would not ensure systems operate safely and ethically in all circumstances. Rather, legal alignment would serve as a *lower bound for safe and ethical AI; it is necessary, but not sufficient.* Approaches to alignment that extend beyond the reach of law could, for example, pursue more ambitious goals like ensuring AI systems support users' long-term health and well-being (Kirk et al., 2025). For the avoidance of doubt, however, progress on legal alignment remains critical given the current status quo in which AI systems are not specifically designed to respect the law (O'Keefe et al., 2025) and have been shown to engage in conduct that, if taken by a human, would be illegal (Scheurer et al., 2024). At the same time, there is a risk that overly rigid legal alignment may itself be undesirable, given that strict compliance with law may sometimes be unjust or harmful (Rawls, 1999). Some violations of law, particularly minor infractions, can be explicitly excused, justified, or forgiven (Minow, 2019), as with necessity defenses (American Law Institute, 1962). Violations of law can sometimes even be morally or socially desirable, as in the case of certain acts of civil disobedience (King, 1963). Seen in this light, the "resistibility" of law is a feature, not a bug (Lazar, 2025). AI systems that *never* resist law or contest entrenched interpretations of law would present new, perhaps even thornier, challenges.

**Should AI systems give effect to laws that are unjust or oppressive?** There is a longstanding debate over whether unjust or immoral laws can be laws at all (Hart, 1958; Fuller, 1957; Raz, 1975; Finnis, 1980; Dworkin, 1986), and whether citizens are morally obligated to comply with laws authored by an illegitimate authority (Ladenson, 1972; Edmundson, 1998). While we do not seek to resolve these contentious issues here, legal alignment forces a confrontation with the question of whether AI systems should be designed to follow such laws. For example, how should legal alignment contend with laws that support genocide (Lemkin, 1944; Arendt, 1963), slavery (Cover, 1975), or racial discrimination (Dyzenhaus, 2010)? The answer in such cases must clearly be that a robustly aligned AI system will *not comply with such laws.* But, in other cases, the answer may be more ambiguous (O'Keefe et al., 2025), such as where the law—including judicial decision-making—is not explicitly immoral but rather reflects or amplifies existing social and economic inequalities or power disparities (Bell, 1980; Unger, 1983; Kennedy, 1991; Moyn, 2024). For instance, should legal alignment uphold tax laws that are extractive and harm a majority of the population but the legality of which remains unchallenged? What about tax laws that primarily harm a politically disempowered minority that cannot effectively challenge those laws through democratic processes (Ely, 1980)? These questions are

---

[2]This can be seen in free speech protections, including under the First Amendment in the United States, which may operate to preclude laws imposing certain restrictions on AI systems (Sunstein, 2024; Salib, 2024).

particularly consequential given that AI systems might not only mirror, but *magnify*, existing legal biases. Admittedly, addressing such concerns by designing AI systems to selectively choose which laws to follow presents many risks. Such discretion could exacerbate legal ambiguity, undermine the universal and equal application of law, and, in time, erode the rule of law itself. While this challenge is not unique to legal alignment and arises, for example, in the exercise of prosecutorial discretion and judicial review (Mashaw et al., 2025), the design of AI systems presents new questions. One potential response is to align AI systems with universal human rights enshrined in international law, including where domestic law may violate such rights (Prabhakaran et al., 2022; Bajgar & Horenovsky, 2023; Samway et al., 2025; Maas & Olasunkanmi, 2025).

## 5.2 Application and edge cases

**Can laws created for humans and human organizations be productively applied to AI systems?**
Designing AI systems to comply with laws that were created to govern human beings and human organizations may be inadequate in the case of actions that are harmless when taken by humans but socially noxious when taken by AI systems that exhibit superhuman intellect, speed, or scale (Morris et al., 2024; Hammond et al., 2025). For example, large numbers of sophisticated AI agents could learn to overcome governance mechanisms designed to prevent manipulation of financial markets by human actors and organizations (Wang & Wellman, 2020). Clearly, existing laws were not generally designed to contend with micro-decisions and actions of billions of AI agents (Gabriel et al., 2024; 2025), let alone organizations and institutions comprised of such agents (Hadfield & Koh, 2025; Tomasev et al., 2025). Another problem concerns the fact that most laws are premised upon the specific capacities and constraints of humans (Simon, 1997) and developed in anticipation of only partial enforcement (Becker, 1968). Because AI systems are not necessarily subject to the same constraints as humans, absolute compliance or perfect enforcement may become practically feasible, but remain socially undesirable (Zittrain, 2008; Brownsword & Yeung, 2008). For example, an autonomous vehicle that perfectly complies with all traffic laws may disrupt established social practices that the public and lawmakers (implicitly) endorse (e.g., breaking the speed limit in a health-related emergency). Finally, many areas of existing law invoke human-centric concepts such as intent and *mens rea* that cannot be straightforwardly applied in the context of AI systems (Nerantzi & Sartor, 2024; Hendrycks, 2024). Tackling these challenges will require both technical work in designing legally aligned AI systems and, possibly, amendments to the law itself in response to the emergence of a new class of non-human actors and organizations.

**What interventions can support AI systems obeying both the letter and spirit of the law?**
AI systems might learn to comply with the formal expression of legal rules but ignore or violate their underlying purpose (Skalse et al., 2022). A failure to act in accordance with background norms, practices, and conventions could be harmful and undermine the prosocial rationale for legal alignment. One approach to resolving this issue involves designing AI systems not only to comply with the substantive content of law (O'Keefe et al., 2025), but to engage in accepted modes of legal reasoning (Caputo, 2025) or possibly adopt an "internal point of view" (Hart, 2012; Shapiro, 2006) whereby AI systems "accept" law as a practical standard to govern their actions, rather than simply seek to avoid legal sanctions (Austin, 1832; Holmes, 1897). This deeper engagement with law, which could be premised on recognizing the legitimacy of legal institutions and procedures (Tyler, 2006), will be critical to ensuring AI systems do not creatively skirt or abuse legal rules (Schneier, 2021), or exploit "legal zero-days," that is, previously undiscovered vulnerabilities in legal frameworks (Sadler & Sherburn, 2025). This approach also finds support in codes of conduct that, for example, prohibit lawyers from making frivolous or abusive claims (American Bar Association, 2020) and preclude judges from expounding absurd statutory interpretations (U.S. Supreme Court, 1868). Meta-rules like these could potentially be adapted to support the legal alignment of AI systems, guiding them to respect both the spirit and letter of the law. One of the main bottlenecks to developing such approaches is the legal reasoning capabilities of AI systems (see Section 2.2). Despite recent improvements (Zheng et al., 2025; Han et al., 2025; Posner & Saran, 2026b), the limitations of current AI systems' legal reasoning capabilities (Doyle & Tucker, 2025; Pruss & Allen, 2025; Purushothama et al., 2026) will need to be overcome in order to design AI systems that understand and comply not only with the law's formal expression but also its underlying purpose.

**How will the participation of AI systems in lawmaking affect legal alignment?** The prospect of AI systems participating in the production of law is growing, whether through generating legal texts

(Wilf-Townsend & Tobia, 2025) such as legislation (Sanders & Schneier, 2025) and even constitutions (Albert & Frazier, 2027), engaging in legal interpretation (Hoffman & Arbel, 2024; Grimmelmann et al., 2025), or rendering judicial opinions (Choi, 2025; Waldon et al., 2025). These developments pose significant challenges for legal alignment (Kolt, 2026). First, law's institutional legitimacy may be undermined if legal rules and principles are no longer developed through human processes of participation and decision-making (Habermas, 1996; Pasquale, 2019). Second, law's legitimacy may be challenged if AI systems fail to fulfill procedural requirements, such as demands for transparency and public explanation, that are key to ensuring accountability and democratic responsiveness (Coglianese & Lehr, 2017). Third, to the extent AI systems shape the content of law that, in turn, governs AI systems, there could emerge a circular process in which these (artificial) subjects of law, in effect, *write their own law*, while lacking the legitimate authority to do so. In addition to blunting the utility of legal alignment in retaining control over AI systems, this process could also erode or distort the rule of law. Similar phenomena can be seen in cases of regulatory capture (Dal Bó, 2006; Carpenter & Moss, 2013) and "legal endogeneity," whereby those actors that the law seeks to control end up controlling the law (Edelman, 1992; 2016). One potential response is to circumscribe the role of AI systems in lawmaking (Kleinberg et al., 2018; Engstrom & Ho, 2020) and ensure that humans retain the ability to make consequential legal decisions (Zanzotto, 2019; Crootof et al., 2023) and, where necessary, intervene in the lawmaking activities of AI systems. The effectiveness and appropriateness of this response could, however, change with the emergence of new perspectives on the role of AI in society (Salib & Goldstein, 2025b;a; Chesterman, 2026; Leibo et al., 2025).

### 5.3 Tradeoffs and future outlook

**Will legal alignment preclude or hamper valuable AI applications?** The implementation of legal alignment could prove costly and come at the expense of societally beneficial AI applications. Conducting rigorous legal alignment evaluations, intervening in the design of AI systems, and complying with associated governance frameworks could impose substantial costs on developers, deployers, and users. Such costs would comprise an "alignment tax" (Askell et al., 2021), that is, the development of legally aligned systems would be subject to additional technical, financial, and procedural burdens relative to other systems that are not legally aligned. This perspective, however, is incomplete. For example, whether legal alignment degrades the performance of AI systems is an open empirical question. Like other alignment methods, legal alignment could potentially *improve* the capabilities of AI systems (Christiano et al., 2017; Ouyang et al., 2022). In particular, AI systems that understand and operate in accordance with law may be especially valuable in high-stakes domains, such as healthcare and finance (Henderson et al., 2024; Hui et al., 2025). In addition, by providing assurances that AI systems comply with the law, legal alignment could reduce the prospects of legal liability for key stakeholders, including developers, deployers, and users (Ayres & Balkin, 2024; Kolt, 2025; Williams et al., 2025b). If this were the case, legal alignment would not comprise an "alignment tax," but rather an "alignment subsidy" that bolsters the performance and practical feasibility of using AI systems, especially in safety-critical applications.

**Could the measurement of legal alignment be gamed or exploited?** While evaluating the legal compliance of AI systems in simple cases such as overt violations of law may be relatively straightforward, developing robust tests to detect more subtle instances of legal misalignment will be difficult. The problem is exacerbated by Goodhart's Law: "when a measure becomes a target, it ceases to be a good measure" (Goodhart, 1975; Strathern, 1997). Developers seeking to improve the performance of AI systems on legal alignment benchmarks may, rather than design systems to uphold core legal principles, inadvertently steer AI systems to violate the law in hard-to-detect ways. Such systems—characterized by *deceptive legal alignment*— could "hack" the law by discovering and exploiting loopholes in legal frameworks (Schneier, 2021; O'Keefe et al., 2025). This, however, would not necessarily differ substantially from lawyers' run-of-the-mill exploitation of legal loopholes to zealously advance their clients' interests (Llewellyn, 1960; Schauer, 2009). More broadly, these measurement challenges are not unique to legal alignment, but implicate many metrics designed to evaluate AI systems (Thomas & Uminsky, 2020; Skalse et al., 2022). One important mitigation in this case is to complement legal alignment benchmarks with dedicated red-teaming efforts that specifically target scenarios not captured by benchmarks, or scenarios for which benchmark results could be misleading (Feffer et al., 2024). In addition, it is possible that legally aligned AI systems could be used to "penetration-test"

and "patch" loopholes in existing law, as well as pilot new legal mechanisms and institutions tailored to address the anticipated affordances of more advanced AI technology (Cuéllar & Huq, 2022).

**Can legal alignment scale to AGI and superintelligence?** Predicting whether legal alignment will succeed in the context of uncertain and contentious future developments is necessarily speculative (Kokotajlo et al., 2025; Narayanan & Kapoor, 2025). Reasoning about the nature, timing, and impact of artificial general intelligence ("AGI") or superintelligence (Morris et al., 2024; Hendrycks et al., 2025) is fraught (Blili-Hamelin et al., 2025). Nevertheless, given the potential stakes of these developments, and the fact that AI developers and policymakers will *in any event* need to make choices concerning the design, deployment, and governance of advanced AI, it is important to inquire whether legal alignment will be effective in the face of systems whose capabilities broadly match or surpass those of humans. There are several reasons for optimism. First, law has a track record of governing increasingly complex activities and actors, such as multinational corporations and government bureaucracies (Muchlinski, 2021; Mashaw et al., 2025). Second, legal data and methods could scale with improvements in AI such that legal alignment continues to remain technically feasible (Nay, 2022; Boeglin, 2026). Third, human-level or superhuman AI systems may help support the implementation of legal alignment, whether by constraining the reasoning and decision-making of these systems (Caputo, 2025) or protecting the underlying laws that guide their behavior (O'Keefe et al., 2025). Of course, these are hopeful predictions. Practical progress will depend on the efforts of researchers and policymakers to iteratively develop and adapt the field of legal alignment as AI systems continue to advance and transform society.

---

**Key takeaway**: Legal alignment faces a variety of open research questions, which primarily concern: (1) the nature and content of law itself, including law's ambiguity and potential for abuse and exploitation; (2) the appropriateness of existing human law for governing computational systems with vastly different characteristics and affordances; (3) costs and tradeoffs associated with implementing legal alignment in practice, as well as the potential efficacy of legal alignment in the face of more advanced AI technologies.

---

## 6    Conclusion

Law offers an underexplored set of rules, principles, and methods for designing safe and ethical AI. Drawing on the institutional legitimacy of law in democratic societies, legal alignment describes a range of roles that legal rules and structures can play in reshaping the design of AI systems to address growing safety and governance concerns. While legal alignment is not a catch-all solution for the many challenges arising from AI, it is both independently important and supportive of complementary alignment research programs. To guide the emerging field of legal alignment, we outline several core areas of focus: using the content of legal rules and principles to steer the behavior of AI systems, leveraging methods of legal reasoning and interpretation to constrain how AI systems make decisions, and harnessing time-tested legal concepts as structural blueprints for tackling problems of alignment. Each of these areas presents new conceptual questions, empirical challenges, and opportunities for technical and institutional innovation. As legal scholars, computer scientists, and researchers spanning multiple disciplines, we look forward to collaborating on this ambitious and pressing agenda.

## Broader Impact Statement

This paper surveys the emerging field of legal alignment and advocates for designing AI systems to operate in accordance with legal rules, principles, and methods. This research agenda can contribute to the creation of safer and more ethical AI. To avoid potential confusion, we wish to explicitly state that legal alignment is not a substitute for legal regulation, particularly regulation that imposes liability on actors responsible for harms arising from the decisions or actions of AI systems. The fact that an AI system is designed to comply with the law does not absolve its developer or user of responsibility. While we suggest that legal alignment will likely reduce the incidence of real-world harms from AI systems, there will nevertheless be circumstances in which developers and/or users should be held legally liable.

## Acknowledgements

For helpful comments and suggestions, we thank Doni Bloomfield, Alan Chan, David Duvenaud, Neel Guha, Martha Minow, Tim Rudner, Tan Zhi Xuan, and participants in the Inaugural Roundtable on AI Safety Law at the University of Alabama Law School and the Sociotechnical AI Safety Retreat at the Australian National University. The Hebrew University Governance of AI Lab and this research are supported by the Israel Science Foundation (Grant No. 487/25), Survival and Flourishing Fund, and Coefficient Giving.

## Appendix – Glossary of Legal Terms

*The following glossary defines key legal terms and concepts used in the paper. It is intended to assist readers without prior legal training or expertise.*

Definitions are adapted from those contained in the Wex Legal Dictionary and Encyclopedia published by Cornell Law School's Legal Information Institute.

**Agency law.** Agency law controls relationships between agents and principals. A principal-agent relationship is created when the agent is given authority to act on behalf of the principal. An agreement made by an agent is binding on the principal so long as the agreement was within the authority actually granted to the agent or reasonably perceived by a third party.

**Canons of construction**. Canons of construction are a system of rules or maxims that are used to interpret legal instruments such as statutes, providing predictable means of resolving ambiguities.

**Conflict of laws**. Conflict of laws refers to a difference between the laws of two or more jurisdictions with some connection to a case, such that the outcome depends on which jurisdiction's law will be used to resolve each issue in dispute.

**Due process**. The legal framework stipulating that all levels of government must operate within the law ("legality") and provide fair procedures.

**Fiduciary duty**. A legal obligation bestowed upon a person (called a "fiduciary") who has been given the authority to act on behalf of another person or entity. A fiduciary relationship exists whenever one party explicitly or sometimes implicitly places trust and confidence in another and the other party accepts responsibility to act on their behalf. This obligation requires fiduciaries to act in the best interests of that person, and not for their own personal gain.

**Formalism**. An approach to jurisprudence that emphasizes the discovery of legal principles through logical analysis, and the application of those principles to the facts of a case. Formalists believe that by applying a consistent set of legal rules to a given case, sound legal decisions will be the outcome of logical deduction.

**Legal person**. A human or a non-human legal entity that is treated as a person for legal purposes. A legal person is capable of engaging in all usual legal business that a real person can participate in, such as suing, being sued, owning property, and entering into contracts.

**Mens rea**. The state of mind statutorily required in order to convict a particular defendant of a particular crime. Establishing the mens rea of an offender, in addition to the actus reus (physical elements of the crime) is usually necessary to prove guilt in a criminal trial. The prosecution typically must prove beyond reasonable doubt that the defendant committed the offense with a culpable state of mind.

**Originalism**. A theory of interpreting legal texts holding that a text in law, especially the U.S. Constitution, should be interpreted as it was understood at the time of its adoption.

**Precedent**. A court decision that is considered an authority for deciding subsequent cases involving identical or similar facts, or similar legal issues. Precedent is incorporated into the doctrine of stare decisis and requires courts to apply the law in the same manner to cases with the same facts.

**Purposivism**. A legal theory that a court's statutory interpretation should reflect the statute's original purpose.

**Textualism**. A method of statutory interpretation that asserts that a statute should be interpreted according to its plain meaning and not according to the intent of the legislature, the statutory purpose, or the legislative history.

**Tort**. An act or omission that gives rise to injury or harm to another and amounts to a civil wrong for which courts impose liability.

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
