# OpenReview forum: "Legal Alignment for Safe and Ethical AI"
_TMLR — Accepted by TMLR_

### Review · Reviewer_2QNk · 2026-02-28

**Summary Of Contributions:**

Summary:
This paper proposes legal alignment as a complementary research agenda within AI alignment: using (1) the content of legal rules and principles as normative targets for AI system design; (2) methods of legal interpretation/reasoning to guide AI decision-making; and (3) legal concepts as blueprints for building reliable, trustworthy, and cooperative AI systems. Beyond defining the concept, the paper motivates legal alignment via law's legitimacy, its granularity and mechanisms for handling disagreement, and its relevance to mitigating risks from increasingly agentic systems. It then sketches an implementation roadmap organized around (i) empirical evaluations of legal (mis)alignment; (ii) technical interventions at different points in the ML pipeline (data, post-training, scaffolding/guardrails, tool use); and (iii) institutional frameworks for disclosure, oversight, and enforcement. The paper closes by enumerating open questions about the nature of law, edge cases, and longer-run tradeoffs.


Strengths:
- The paper identifies an important gap between AI alignment research and the long-standing societal machinery for specifying and contesting norms: the legal system. This is a timely topic as more agentic-AI systems are built (and targeted to make decisions for humans).
- The paper is generally well-written. The three ``pathways'' (legal content, interpretive methods, structural blueprints) are a useful organizing scheme that helps separate normative targets from reasoning processes and from institutional design.
-  The ``implementation'' section provides a concrete taxonomy of evaluation approaches and intervention sites that can be used to plan projects and benchmark progress.


Weaknesses:
- Limited concrete technical content. The paper is primarily an agenda/position piece and offers few worked examples, toy systems, or case studies that demonstrate how legal alignment would change the behavior of real models or agents in practice.

-  Key notions like legal compliance and legal misalignment can be hard to evaluate for stochastic, tool-using agents.

**Audience:**

Yes

**Audience Explanation:**

TMLR readers working on alignment, safety evaluation, agentic systems, and responsible ML should find the paper useful as (i) a conceptual bridge to law, and (ii) a source of research questions for building and evaluating AI systems that must operate under real-world constraints.

**Broader Impact Concerns:**

I don’t see any broader impact concerns.

**Claims And Evidence:**

Yes

**Claims Explanation:**

My answer is yes but with reservations.

For a conceptual paper, the core claims are supported primarily by structured argumentation and extensive citations to work on AI alignment, AI governance, and legal theory. The manuscript is generally careful about separating (i) descriptive claims about current alignment practices, (ii) normative claims about legitimacy and rule-of-law values, and (iii) forward-looking claims about what legal alignment could enable. Several practically important claims are mostly defended by plausibility arguments. In my opinion, adding at least a few concrete examples or preliminary evaluation results would make the evidence base more compelling.

**Requested Changes:**

- Add at least one worked example / mini-case study and show how each of the three pathways would concretely alter the system design and evaluation.

- Consider adding a short table mapping each pathway to (i) example tasks/domains, (ii) candidate evaluation methods, and (iii) plausible technical levers (prompting, retrieval, training signals). This would help readers quickly translate the agenda into project plans.

---

> ### Author Response · Authors · 2026-04-06
> **Thanks + response**
>
> We are thankful for your helpful comments and suggestions. We were glad to hear that you believe the paper identifies an important gap in AI alignment research and found it useful for TMLR readers and generally well-written. **We have uploaded a revised version of the paper with changes made in blue that we hope address your concerns.**
>
> ---
>
> ## 1. Concrete case studies
>
> > The paper is primarily an agenda/position piece and offers few worked examples, toy systems, or case studies … Add at least one worked example / mini-case study and show how each of the three pathways would concretely alter the system design and evaluation.
>
> **We agree that concrete case studies would improve the paper, and have added these.** Our contribution is primarily a survey paper that systematizes research in the emerging field of legal alignment and outlines directions for future technical and governance research. That being said, we strongly agree that concrete case studies will improve our paper.
>
> Thanks to your suggestion (as well as complementary suggestions by Reviewer zHAB), our contribution has now been strengthened by the inclusion of concrete worked examples that illustrate how legal alignment could be designed for, evaluated, and enforced in real-world systems.
>
> **Action 1:** We have added a new Section 4.4 containing concrete case studies illustrating how the three legal alignment pathways operate in practice. (This is also in line with Reviewer Njbe’s suggestions.)
>
> ---
>
> ## 2. Table operationalizing legal alignment pathways
>
> > Consider adding a short table mapping each pathway to (i) example tasks/domains, (ii) candidate evaluation methods, and (iii) plausible technical levers
>
> **Action 2:** We have added a table in the new Section 4.4. The table illustrates the practical role of empirical evaluation, technical interventions, and institutional frameworks (Sections 4.1, 4.2, and 4.3) across each of the three legal alignment pathways (Section 2.1), thereby connecting the conceptual research agenda to its concrete, real-world application. The table also complements the concrete case studies (as per item 1 above).
>
> ---
>
> ## 3. Challenges in evaluating legal alignment of agents
>
> > Key notions like legal compliance and legal misalignment can be hard to evaluate for stochastic, tool-using agents.
>
> **We agree that evaluating legal alignment is especially difficult for agentic systems.** Several characteristics of contemporary AI agents compound this difficulty, including LLM-based agents’ nondeterministic operation and their ability to access and use external tools and resources. The former, for example, requires that legal alignment evaluations include testing LLM-based agents under different temperatures (which control the stochasticity of model outputs); the latter, meanwhile, requires that legal alignment evaluations include different experimental conditions under which agents are provided access to different tools that may improve legal alignment, such as access to legal documents and legal databases.
>
> **Action 3:** We have expanded the discussion of agentic evaluation challenges in Section 4.1 to encompass additional considerations, including considerations specific to evaluating the legal alignment of agents (as opposed to agentic evaluations more broadly).
>
> ---
>
> ## Any additional feedback?
>
> **Thank you again for your detailed and constructive feedback. We believe and hope these changes address your core critiques of our paper. We would be happy to know what other changes we can make to further improve the paper, so please let us know if you have additional feedback. Thank you!**

---

### Review · Reviewer_Njbe · 2026-03-08

**Summary Of Contributions:**

This paper introduces legal alignment as an emerging interdisciplinary field that leverages legal rules, principles, and methods to address the normative and technical challenges of AI alignment, filling a critical gap in existing AI alignment research that has largely overlooked law as a foundational resource for safe and ethical AI design. The core contributions are fourfold:

Definition and Context: Establishes legal alignment as the design of AI systems to operate in accordance with legal frameworks, distinguishing it from AI legal regulation (which governs human/ organizational actors) and grounding it in parallels between legal reasoning and AI alignment challenges (e.g., principal-agent problems, ambiguous rule interpretation).

Rationale: Articulates four core justifications for legal alignment—institutional legitimacy/process of law, structural features of law, responsiveness to AI safety/governance risks, and practical/societal feasibility—and details how law’s transparency, accountability, and context-sensitivity address key limitations of dominant AI alignment approaches (e.g., opaque developer-led specifications, failure to balance plural societal values).

Three Core Research Pathways: Outlines the foundational directions of legal alignment:

Pathway 1: Using legal rules/principles as normative content for AI alignment (e.g., designing AI to comply with jurisdiction-specific laws as if it were a human actor).

Pathway 2: Adapting legal interpretation/theory to guide AI reasoning (e.g., analogical reasoning from legal precedent, textualist canon interpretation for ambiguous safety specs).

Pathway 3: Harnessing legal concepts/institutions as a structural blueprint for AI alignment (e.g., agency law, fiduciary duty to address reliability/trust in AI systems).

Implementation and Open Questions: Provides a practical roadmap for legal alignment (empirical evaluations, technical interventions, institutional frameworks) and identifies critical open research questions across the nature of law, real-world application edge cases, and tradeoffs for future AI (including AGI/superintelligence).

Additionally, the paper clarifies key distinctions (legal alignment vs. AI regulation, legal compliance vs. ethical AI) and emphasizes that legal alignment acts as a critical lower bound for safe AI (not a panacea) that complements—rather than replaces—existing alignment approaches (e.g., Constitutional AI, RLHF).

**Audience:**

Yes

**Audience Explanation:**

TMLR’s interdisciplinary audience (machine learning researchers, AI safety scientists, computer scientists, and social scientists focused on AI) will find the findings highly relevant and valuable, with appeal across multiple subcommunities:

Core ML/AI Alignment Researchers

TMLR is a leading venue for AI alignment research, and this paper introduces a novel, principled framework to address longstanding limitations of dominant alignment approaches (e.g., opaque developer specs, failure to incorporate societal values). The three legal alignment pathways provide concrete new directions for technical research (e.g., legal interpretation for AI reasoning, case-based alignment from legal precedent) that complement existing work on Constitutional AI, RLHF, and pluralistic alignment.

AI Safety and Agentic AI Researchers

The paper addresses critical AI safety risks (systemic multi-agent harm, algorithmic collusion, autonomous hacking) and provides a legal framework to mitigate these risks—an area of intense focus for TMLR’s audience. The discussion of legal alignment as a tool to prevent illegal AI behavior (e.g., insider trading, fraud) and protect the rule of law is directly relevant to research on safe agentic AI and frontier model governance.

Interdisciplinary ML/ Social Science Researchers

TMLR increasingly publishes work at the intersection of ML and social science (ethics, governance, policy). This paper bridges computer science and law, providing a roadmap for interdisciplinary collaboration—an area of growing interest for TMLR authors and readers. The discussion of institutional frameworks (e.g., AI model registration, legal alignment certification) is also highly relevant to researchers focused on AI policy and governance.

Industry ML Practitioners

While TMLR is an academic venue, many readers work in industry (OpenAI, Anthropic, Meta, Google) on alignment and safety for deployed LLMs/frontier models. The paper’s practical implementation roadmap (empirical evaluations, technical interventions, institutional disclosure) provides actionable guidance for designing legally compliant AI systems—an urgent priority for industry given growing global AI regulation (EU AI Act, U.S. state laws).

Even for researchers focused on purely technical alignment (e.g., formal verification, RLHF optimization), the paper offers a new lens for grounding technical work in societal norms and legal legitimacy— a critical consideration for translating ML research to real-world deployment.

**Claims And Evidence:**

Yes

**Claims Explanation:**

The paper’s claims are broadly supported by rigorous, multi-disciplinary evidence and clear reasoning, with a few minor gaps in empirical validation for emerging technical claims.

Key strengths and caveats:

Strengths:

Foundational Legal and AI Alignment Scholarship: The paper synthesizes an extensive body of research from law (jurisprudence, regulatory theory), AI alignment (RLHF/RLAIF limitations, Constitutional AI), and AI safety (systemic risks, agentic AI harms), with over 400 citations to peer-reviewed work, policy documents, and leading industry reports (e.g., OpenAI/Anthropic model specs, EU AI Act, California SB-53).
Empirical Limitations of Current AI Alignment: Convincingly documents failures of dominant alignment approaches (e.g., untruthful content, bias, sycophancy, jailbreaks) with citations to recent empirical studies of state-of-the-art LLMs (ChatGPT, Claude, Gemini), providing clear evidence for why a new legal alignment framework is needed.

Legal Institutional and Structural Evidence: Draws on well-established legal theory (Hart, Dworkin, Raz) and real-world legal mechanisms (conflict of laws, fiduciary duty, administrative rulemaking) to justify how law addresses AI alignment’s normative and technical gaps—evidence that is accurate and widely accepted in legal scholarship.

Practical Technical Precedents: Cites existing integration of legal resources in AI development (e.g., legal texts in pre-training datasets, Llama Guard’s legal hazard taxonomies, Anthropic’s constitutional incorporation of human rights law) to support the feasibility of legal alignment interventions.

Minor Gaps:

Limited Empirical Data for Legal Alignment Evaluations: The paper proposes novel empirical evaluation methods for legal alignment (e.g., agentic evaluation environments, legal compliance benchmarks) but notes that these are emerging (few existing studies). While this is acknowledged openly, it leaves some technical claims (e.g., the efficacy of legal interpretation for AI reasoning) as theoretically compelling but not yet empirically validated at scale.

Case Studies of Legal Alignment in Practice: The paper references policy support (CA SB-53, NY RAISE Act) but lacks concrete case studies of deployed AI systems designed with legal alignment in mind. This is a minor gap, as legal alignment is an emerging field with few real-world implementations to date.

Cross-Jurisdictional Legal Complexity: The paper addresses jurisdictional challenges (e.g., which laws apply to global AI systems) but provides limited evidence for how *conflict-of-laws principles* can be technically implemented in AI systems—this is a technical gap, not a flaw in the core claim.

Overall, the claims are convincing and accurate for a foundational paper in an emerging field; the gaps are openly acknowledged and framed as open research questions, which is appropriate for a TMLR submission.

**Requested Changes:**

To strengthen the paper for TMLR, we propose minor revisions (mostly clarifications and additions) that address the empirical gaps noted above and improve the accessibility of legal concepts for a primarily computer science audience. All changes are aligned with the paper’s core argument and emerging field focus:

A. Accessibility for Computer Science/ML Audience

Glossary of Key Legal Terms: Add a brief (1–2 page) appendix with definitions of core legal concepts (e.g., textualism, fiduciary duty, conflict of laws, mens rea, rule of law)—many TMLR readers will not have legal training, and this will reduce the cognitive load of engaging with the paper’s legal scholarship.

Simplified Legal Alignment Technical Pipeline: Add a figure that visualizes the three legal alignment pathways (1–3) and their integration into the standard AI development pipeline (pre-training → post-training → deployment). This will make the technical implications of legal alignment more concrete for ML researchers.

B. Technical Clarification of Cross-Jurisdictional Implementation

Section 2.1 (Pathway 1) Revision: Add a short subsection that outlines technical approaches for implementing conflict-of-laws principles in AI systems (e.g., geolocation-based legal rule retrieval, jurisdictional prompt engineering, modular legal knowledge bases). This addresses the gap in technical guidance for cross-jurisdictional legal alignment and makes the paper more actionable for ML engineers.

---

> ### Author Response · Authors · 2026-04-06
> **Thanks + response**
>
> We are thankful for your helpful comments and suggestions. We were glad to hear that you found our analysis “convincing and accurate” and consider our contribution to be “a foundational paper in an emerging field” that is appropriate for TMLR. **We have uploaded a revised version of the paper with changes made in blue that we hope address your concerns.**
>
> ---
>
> ## 1. Empirical data on legal alignment evaluations
>
> > The paper proposes novel empirical evaluation methods for legal alignment … but notes that these are emerging (few existing studies).
>
> **As a survey paper, our contribution aims to highlight this challenge and propose how it can be addressed.** We strongly agree that legal alignment currently lacks sufficient empirical evaluation. By discussing the limited existing empirical studies (Sections 2.2 and 4.1) and exposing their shortcomings, we aim to highlight this gap and, thereby, motivate AI researchers to design and conduct evaluations to fill that gap. Section 4.1's discussion of evaluation methodology is deliberately forward-looking: it highlights evaluation techniques that should be (but have not yet been) applied to evaluating legal alignment. Structurally, our contribution is similar to other survey papers published in TMLR (e.g., [Reuel, Bucknall et al., 2025](https://openreview.net/forum?id=1nO4qFMiS0); [Casper et al. 2026](https://openreview.net/forum?id=8QyGLnFkzc)) that highlight gaps in current research and propose avenues for future empirical work.
>
> **Action 1:** We have revised Section 4.1 to clarify that (a) because legal alignment evaluations are still a nascent research area, existing studies remain relatively limited; and (b) our discussion is deliberately forward-looking as it aims to motivate and outline directions for future work.
>
> ---
>
> ## 2. Case studies of legal alignment in practice
>
> > The paper references policy support (CA SB-53, NY RAISE Act) but lacks concrete case studies of deployed AI systems designed with legal alignment in mind.
>
> **A key goal of our paper is to illustrate the current lack of legal alignment in practice.** We agree with the reviewer’s observation that, despite nascent policy support for legal alignment, few deployed AI systems are explicitly designed to incorporate legal alignment. Describing this finding is, in our view, itself a valuable contribution. Nevertheless, we appreciate the reviewer’s suggestion to include concrete case studies, particularly to illustrate how legal alignment could be designed for, evaluated, and enforced in practice.
>
> **Action 2:** We have (a) clarified in Section 2.1 that the current lack of institutional support for legal alignment in practice is itself a noteworthy research finding; (b) added a new subsection (Section 4.4) with concrete case studies that serve as worked examples of implementing legal alignment in real-world settings. (This is also in line with suggestions from Reviewer 2QNk.)
>
> ---
>
> ## 3. Clarification of cross-jurisdictional implementation
>
> > Add a short subsection that outlines technical approaches for implementing conflict-of-laws principles in AI systems
>
> **Action 3:** Thank you for this helpful suggestion. We have added a subsection at the end of Pathway 1 (in Section 2.1) to expand our analysis of cross-jurisdictional and conflict-of-laws issues. The new subsection (a) discusses technical methods for enabling AI systems to determine the applicable jurisdiction; and (b) proposes conducting evaluations to assess systems’ ability to make such determinations in practice.
>
> ---
>
> ## 4. Glossary of key legal terms
>
> > Add a brief (1–2 page) appendix with definitions of core legal concepts
>
> **Action 4a:** We greatly appreciate this suggestion, particularly given that many of TMLR’s readers do not have prior legal expertise or training. Accordingly, we have added an appendix that contains a glossary of key legal terms mentioned in the paper (e.g., originalism, mens rea, torts, agency law).
>
> **Action 4b:** We have also added “takeaway boxes” after Sections 2 through 5 in order to assist AI researchers (without legal training) in gleaning the core insights from our paper.
>
> ---
>
> ## 5. Simplified legal alignment technical pipeline
>
> > Add a figure that visualizes the three legal alignment pathways (1–3) and their integration into the standard AI development pipeline
>
> **Action 5:** This is a very helpful suggestion. We have added this figure in the new Section 4.4 in order to illustrate how the three pathways of legal alignment can be incorporated into existing AI development and deployment practices. As per item 2 above, this figure complements the concrete case studies.
>
> ---
>
> ## Any additional feedback?
>
> **Thank you again for your detailed and constructive feedback. We believe and hope these changes address your core critiques of our paper. We would be happy to know what other changes we can make to further improve the paper, so please let us know if you have additional feedback. Thank you!**

---

### Review · Reviewer_zHAB · 2026-03-25

**Summary Of Contributions:**

This paper introduces and systematizes the concept of "legal alignment" which advocates for using legal rules, principles, and methods to address AI alignment problems. The authors propose three main research directions: (1) designing AI to comply with legal rules, (2) using legal interpretation methods to guide AI reasoning, and (3) leveraging legal concepts to build reliable and trustworthy AI systems. The paper outlines the stages of AI development where law intersects with technology, discusses the difference between legal regulation and legal alignment, and proposes empirical, technical, and institutional pillars for implementation.
The paper tackles a highly relevant and timely topic, making a valuable distinction between legal regulation and legal alignment. The three pathways identified offer a structured way to think about interdisciplinary collaboration between law and computer science. However, the paper suffers from a mismatch with its target audience, leaning heavily toward legal scholars rather than the AI practitioners that TMLR serves. Furthermore, it overclaims its contributions by presenting the definition of legal alignment as novel despite citing existing literature that already defines it. The technical discussions are often shallow, failing to provide actionable methodologies for AI developers (e.g., how to practically enforce legal alignment), and the paper frequently conflates general AI with Generative AI/LLMs.

**Additional Comments:**

Paper Strengths:
- The problem of aligning AI with societal and legal norms is one of the most critical challenges in the field today.
- The paper draws a very significant and useful distinction between legal regulation (imposing external requirements) and legal alignment (internalizing legal reasoning into the system).
- The categorization of the field into three distinct pathways provides a helpful systematization for a highly interdisciplinary challenge.

Paper Weaknesses:
- The narrative is geared more toward legal scholars than AI practitioners, lacking the technical depth expected in a machine learning venue.
- The authors claim to define the core focus of legal alignment, which conflicts with their own citations of recent papers that have already established this concept. Furthermore, the paper claims to address "AI" broadly but almost exclusively focuses on LLMs/GenAI.
- Sections detailing implementation (evaluation, technical interventions, institutional frameworks) are shallow. They tell the reader what should be done (which is often trivial) without providing insights into how it can be achieved technically.
- The paper seemingly argues in favor of legal alignment over legal regulation, downplaying the fundamental necessity of regulating the actors who produce and deploy these systems.

**Audience:**

Yes

**Audience Explanation:**

While the overarching topic of AI alignment is of significant interest to the TMLR community, the current framing of the paper would likely hold marginal interest for this specific audience. The presentation and discussion are heavily biased towards researchers and practitioners in the legal domain. For TMLR readers (who are predominantly AI researchers, developers, and practitioners) the lack of actionable technical guidance, empirical evaluation frameworks, and concrete implementation strategies makes the findings difficult to apply. A journal such as Springer's "Artificial Intelligence and Law" might find a much broader and more engaged audience for the paper in its current state.

**Broader Impact Concerns:**

While the paper itself deals explicitly with the societal and ethical implications of AI (and thus inherently touches on broader impacts), there is a slight concern in its framing. By inadvertently positioning legal alignment (a technical, system-internal solution) as a potential substitute for, or as more important than, legal regulation (holding human actors and corporations accountable), the paper flirts with a narrative that could be used by tech entities to evade regulatory oversight. Adding a brief Broader Impact Statement clarifying that technical legal alignment does not absolve AI developers of their regulatory and legal liabilities would strengthen the work and mitigate this concern.

**Claims And Evidence:**

No

**Claims Explanation:**

Currently, the claims are not sufficiently supported by deep technical evidence or rigorous critical analysis. Because this acts largely as a position or survey paper, the "evidence" relies on the strength of its analytical arguments. However, these arguments frequently remain at a surface level. For instance, when discussing legal resources in AI development, the paper ignores real-world enforcement challenges (such as recent copyright lawsuits involving data collection). Similarly, the discussion on technical interventions points out obvious intervention sites without offering concrete methodologies or architectures (such as neuro-symbolic AI) that would prove these interventions are feasible in practice.

**Requested Changes:**

- The contributions made by the paper are not clearly presented in the abstract. Is it a survey/systematization paper, or a technical contribution offering a new technique to ensure the legal alignment of AI systems? The authors must clarify this.
- The paper builds on well-established legal alignment papers (e.g., Kolt, 2025; Caputo, 2025; O’Keefe et al., 2025; Boeglin, 2026), yet the authors claim the definition of the core focus of legal alignment as one of their own contributions. The notion and definition of legal alignment are already present in the literature and cannot be considered a novel contribution here. However, a systematization or survey of approaches in this field is valuable. The authors should rephrase the introduction to accurately reflect their contribution as a systematization rather than a novel definition.
- The introduction reveals confusion regarding the definition and concept of AI. The authors use AI as a general term but only provide examples relevant to GenAI models (specifically LLMs like ChatGPT, Claude, and Gemini). Legal alignment is crucial across all AI contexts (e.g., avoiding bias in medical AI diagnosing diseases from MRI scans). By focusing solely on LLMs while claiming to address general AI, the authors oversell their contribution and weaken the narrative. The scope should be explicitly narrowed to LLMs, or the examples must be broadened.
- The three pathways presented are valuable but are biased toward legal practitioners. It is difficult to understand how to pursue these pathways during the actual development and deployment of AI technologies. The authors should extend their discussion to provide actionable inspiration and guidance for AI developers.
- The list of AI development stages where law comes into play is interesting but shallow. The authors should delve deeper. For instance, are jurisdiction-specific laws actually enforced during data collection? (Considering Anthropic's recent copyright breach settlement, this is doubtful). Are guardrailing policies working in practice? A more critical discussion of these real-world challenges is needed.
-  The distinction between regulation and alignment is significant, but the paper currently seems to argue in favor of alignment over regulation. This is a risky direction, as imposing requirements on the actors producing and using AI remains fundamental. The authors should restructure the presentation to stress that both aspects are essential and neither replaces the other.
- Section 3 is overly abundant and the points made are quite trivial; it is straightforward to understand why legal alignment is needed and what goes wrong without it. The authors should make this section concise and more grounded. For example, explicitly discuss what happens technically or socially if an AI system is grounded on "illegitimate" rules instead of legitimate ones.
- As the most relevant section for AI practitioners, the presentation of the three "pillars" (empirical evaluation, technical interventions, institutional frameworks) in Section 4 is too shallow. The authors mention general features but do not explain how to design evaluation frameworks or enact technical interventions in practice. For instance, regarding technical interventions (e.g., guided learning or adding legal resources to data), the authors should discuss actionable methodologies, such as neuro-symbolic or symbolic knowledge injection approaches. Adding references like [1] and [2] would be highly valuable here.
- The subsection on institutional frameworks is confusing. How do these frameworks differ from standard legal regulations that impose requirements on actors that produce AI? What would these frameworks look like in practice, and what are the consequences for an AI system that fails to ensure legal alignment?
- In Section 5 the open research questions are limited. The authors should discuss how a reactive-by-design framework like the law can adapt to a rapidly mutating AI landscape. AI systems might look completely different in 10 to 20 years, creating a bottleneck where outdated laws struggle to govern cutting-edge technology. This is a vital challenge worth discussing.
- The fundamental findings are not immediately clear to the reader. Adding quick "takeaway boxes" at the end of specific sections or subsections would help readers immediately grasp the core findings.
- The paper contains a few typos and grammatical errors (e.g., "sycophantic"). A thorough proofreading pass is required.

[1]. Marra, Giuseppe, et al. "From statistical relational to neurosymbolic artificial intelligence: A survey." Artificial Intelligence 328 (2024): 104062.

[2]. Ciatto, Giovanni, et al. "Symbolic knowledge extraction and injection with sub-symbolic predictors: A systematic literature review." ACM Computing Surveys 56.6 (2024): 1-35.

---

> ### Author Response · Authors · 2026-04-06
> **Thanks + response (part 1)**
>
> We are thankful for your helpful comments and suggestions. We were glad to hear that you found that the paper provides a helpful systematization of a highly relevant and timely topic. **We have uploaded a revised version of the paper with changes made in blue that we hope address your concerns.**
>
> ---
>
> ## 1. Survey/systematization paper
>
> > The authors should rephrase the introduction to accurately reflect their contribution as a systematization
>
> **We agree with and appreciate this characterization of our contribution.** Specifically, our paper surveys and taxonomizes the main aspects of legal alignment (Section 2), the conceptual rationale underpinning the field (Section 3), approaches to the technical implementation of legal alignment (Section 4), and open questions facing the field (Section 5). Such surveys, including those previously published in TMLR (e.g., [Casper et al., 2023](https://openreview.net/forum?id=bx24KpJ4Eb); [Anwar et al., 2024](https://openreview.net/forum?id=oVTkOs8Pka); [Reuel, Bucknall, et al., 2025](https://openreview.net/forum?id=1nO4qFMiS0)), are critical to informing technical researchers about emerging research problems and opportunities.
>
> **Action 1:** We have revised the abstract and introduction (Section 1) to clarify the paper’s primary goal of surveying and systematizing the field of legal alignment.
>
> ---
>
> ## 2. Significance and novelty of contribution
>
> > it is straightforward to understand why legal alignment is needed and what goes wrong without it
>
> **We share your perspective that legal alignment is much-needed, but are concerned that others might not.** To date, most AI safety and alignment research has generally overlooked or dismissed the role of law and legal compliance in developing safe and ethical AI systems. For example, there are relatively few technical papers that empirically evaluate the legal compliance of AI systems (all those we were able to identify are cited in Section 4.1), especially relative to the number of papers that evaluate the general legal capabilities of AI systems (cited in Section 2.2) and papers that measure other (non-law) aspects of AI safety (including those cited in Section 1, page 2).
>
> Moreover, we see our survey of legal alignment as particularly crucial and timely given recent steps to relegate the role of law in AI safety. For example, in January 2026, Anthropic removed references to particular legal documents from Claude’s “Constitution”, including the Universal Declaration of Human Rights, replacing them with references to model “character” and “values” ([Anthropic, 2023](https://www.anthropic.com/news/claudes-constitution); [Anthropic, 2026](https://www.anthropic.com/news/claude-new-constitution)). In light of such developments, we see our paper as offering a critical corrective.
>
> **Key parts of our paper are novel.** Although our primary contribution is a survey of this emerging field, several parts of our paper are novel, including (a) the taxonomy comprised of legal alignment pathways 1–3; (b) discussion of legal resources in current AI development and deployment; (c) clarification of the distinction between AI legal reasoning capabilities and legal alignment; (d) the technical research agenda in Sections 4.1–4.4; and (e) many of the open questions discussed in Section 5.
>
> **Research accessibility is a valuable contribution.** In addition to offering the first comprehensive survey of prior literature on legal alignment, our paper aims to make accessible to computer scientists and technical AI researchers legal scholarship that is often lengthy, jargon-heavy, and tailored to legal professionals (e.g., [Kolt, 2025](https://papers.ssrn.com/sol3/papers.cfm?abstract_id=4772956); [Caputo, 2025](https://papers.ssrn.com/sol3/papers.cfm?abstract_id=4800894); [O’Keefe et al., 2025](https://papers.ssrn.com/sol3/papers.cfm?abstract_id=5242643); [Boeglin, 2026](https://papers.ssrn.com/sol3/papers.cfm?abstract_id=5870922)). We see this as an independently valuable contribution. It is also for this reason that we submitted the paper to TMLR, whose primary audience is AI researchers and practitioners.
>
> **Action 2:** We have explicitly clarified in the introduction the goals that our paper aims to accomplish, including (a) offering a detailed presentation of a novel research agenda; and (b) making this field more accessible to computer scientists.

---

> > ### Author Response · Authors · 2026-04-06
> > **Response (part 2)**
> >
> > ## 3. Fit for TMLR audience
> >
> > > the paper suffers from a mismatch with its target audience, leaning heavily toward legal scholars rather than the AI practitioners that TMLR serves
> >
> > **TMLR encourages publishing sociotechnical and technical governance AI research.** Papers published in TMLR include studies that combine methods from different sociotechnical disciplines to address pressing challenges in AI, including the use of LLMs in finance, healthcare, and law ([Chen et al., 2024](https://openreview.net/forum?id=upAWnMgpnH)), responsible AI development ([Longpre et al., 2024](https://openreview.net/forum?id=tH1dQH20eZ)), transparency indices ([Bommasani et al. 2025](https://openreview.net/forum?id=x6fXnsM9Ez)), and governance-related research agendas ([Reuel, Bucknall et al., 2025](https://openreview.net/forum?id=1nO4qFMiS0)). These papers have been awarded certifications, including the “Survey” and “Featured” [certifications](https://jmlr.org/tmlr/editorial-policies.html). We see our paper, which surveys the emerging sociotechnical field of legal alignment, as following a similar path.
> >
> > **Our primary audience is AI researchers and practitioners.** While some of the theoretical legal literature our paper draws upon (e.g., [Kolt, 2025](https://papers.ssrn.com/sol3/papers.cfm?abstract_id=4772956); [Caputo, 2025](https://papers.ssrn.com/sol3/papers.cfm?abstract_id=4800894)) is indeed tailored to legal scholars, our paper explicitly focuses on a different audience: AI researchers and practitioners, most of whom are presumably not familiar with legal alignment. Concretely, Sections 3 and 5 summarize the rationale and open questions in legal alignment in a manner that is accessible to AI researchers, using explanations and terminology that AI researchers will find informative and helpful. In addition, most of Section 4 outlines a technical research agenda comprised of legal alignment evaluations and design interventions that is addressed specifically to AI researchers (and **not** legal scholars, who generally lack the technical skills to pursue such research).
> >
> > **Action 3a:** We have revised the introduction and Section 4 to clarify that our paper primarily addresses AI researchers, in both systematizing the field of alignment and outlining concrete directions for technical research.
> >
> > **Action 3b:** We have added “takeaway boxes” after Sections 2 through 5 in order to assist AI researchers in gleaning the core insights from our paper (see item 10 below).
> >
> > **Action 3c:** We have added an appendix that defines key legal terms and concepts used in the paper, which we believe will significantly assist AI researchers (without prior legal training or expertise) in understanding and engaging with legal alignment.
> >
> > ---
> >
> > ## 4. Technical evidence and critical analysis
> >
> > > the claims are not sufficiently supported by deep technical evidence or rigorous critical analysis
> >
> > **Survey papers like ours outline frameworks and methods for collecting evidence; in addition, we believe our paper contains extensive critical analysis.** TMLR’s [editorial policy](https://jmlr.org/tmlr/editorial-policies.html) explicitly welcomes the publication of “surveys that draw new connections, highlight trends, and suggest new problems in an area” but do not directly contribute deep technical evidence. For example, [Reuel, Bucknall et al. (2025)](https://openreview.net/forum?id=1nO4qFMiS0) outline and critically analyze open problems in AI governance without attempting to offer novel technical findings. We similarly view our contribution as outlining the key challenges in legal alignment, which includes extensively surveying the existing technical literature and identifying where it falls short (e.g., the discussion in Section 4.1 regarding the limited scope of current legal alignment evaluations). In addition, Section 5 offers the most in-depth and comprehensive critical analysis of legal alignment to date; this is especially evident when comparing our account of legal alignment to the more emphatic analyses in [O’Keefe et al. (2025)](https://papers.ssrn.com/sol3/papers.cfm?abstract_id=5242643) and [Boeglin (2026)](https://papers.ssrn.com/sol3/papers.cfm?abstract_id=5870922), which generally advocate particular approaches to legal alignment without devoting significant attention to their shortcomings.

---

> > > ### Author Response · Authors · 2026-04-06
> > > **Response (part 3)**
> > >
> > > ## 5. Defining AI more broadly than LLMs / GenAI
> > >
> > > > The authors use AI as a general term but only provide examples relevant to GenAI models
> > >
> > > **We thank the reviewer for this important comment and agree that legal alignment encompasses other forms of AI.** Our decision to focus primarily on LLMs was based on several considerations, particularly the significant fraction of current technical research that is focused on LLMs, including AI safety and ethics research, as well as the far-reaching societal impacts of LLMs (as we discuss in Section 1). That being said, we strongly identify with the reviewer’s concern that such a focus is too narrow and misses important opportunities to evaluate and implement legal alignment, such as illegal discrimination in medical AI applications and other use cases that do not involve LLMs.
> > >
> > > **Action 5:** We have revised Section 2 to (a) clarify our working definition of AI, indicating that it is broader than GenAI/LLMs; and (b) inserted examples of non-LLM AI applications of legal alignment, including the medical application proposed by the reviewer.
> > >
> > > ---
> > >
> > > ## 6. Legal regulation vs. legal alignment
> > >
> > > > The distinction between regulation and alignment is significant, but the paper currently seems to argue in favor of alignment over regulation.
> > >
> > > **We support both legal regulation and legal alignment, and clarify the different functions they perform.** In Sections 1 and 2.2, we clarify the distinction between legal regulation and legal alignment: the former focuses on regulating actors that develop and deploy AI, while the latter  focuses on integrating law into the design and operation of AI systems themselves. **At no point do we argue that legal regulation is unnecessary or less preferable.** On the contrary, we suggest (on page 3) that “[t]he two fields … are closely related and potentially mutually supportive including because legal regulation can help facilitate legal alignment in practice, such as by enabling researchers to access technical resources required to effectively evaluate and improve the legal alignment of deployed systems.” Section 4.3, which outlines institutional frameworks for implementing legal alignment, essentially illustrates how binding legal regulation is critical to supporting legal alignment.
> > >
> > > **Some legal and regulatory issues are outside the scope of our paper.** We appreciate the reviewer raising the issues of “recent copyright lawsuits involving data collection”, additional “AI development stages where law comes into play”, and “real-world enforcement challenges” concerning other potentially unlawful actions by AI companies. While these issues are of pivotal importance and the subject of both legal and computer science scholarship (e.g., [Henderson et al., 2023](http://www.jmlr.org/papers/v24/23-0569.html); [Sag, 2023a](https://fordhamlawreview.org/wp-content/uploads/2024/03/Vol.-92_Sag-1887-1921.pdf); [2023b](https://houstonlawreview.org/article/92126-copyright-safety-for-generative-ai)), they are outside the scope of our paper, which focuses specifically on “designing AI systems to operate in accordance with legal rules” (page 3), as opposed to broader issues concerning potential legal violations of AI companies.
> > >
> > > **Action 6:** We have clarified in the introduction, Section 2.2, and in the **new Broader Impact Statement** (see item 11 below) that (a) legal regulation can play an instrumental role in facilitating legal alignment; and (b) legal alignment is not a substitute for legal regulation, particularly regulation that imposes liability.
> > >
> > > ---
> > >
> > > ## 7. Technical methodologies
> > >
> > > > the authors should discuss actionable methodologies, such as neuro-symbolic or symbolic knowledge injection approaches
> > >
> > > **We agree that discussing actionable technical methodologies can improve and enrich our paper.** Following the earlier comment regarding GenAI and LLMs, we agree that the discussion of technical interventions in Section 4.2 should be expanded to include additional actionable methodologies, beyond those already discussed in the paper (which are mainly LLM-oriented interventions, such as methods focused on pretraining datasets and post-training alignment techniques). We appreciate the reviewer highlighting important symbolic and neuro-symbolic methods.
> > >
> > > **Action 7:** We have expanded Section 4.2 to include additional technical methodologies, including [Marra et al. (2024)](https://www.sciencedirect.com/science/article/pii/S0004370223002084) and [Ciatto et al. (2024)](https://dl.acm.org/doi/10.1145/3645103) as proposed by the reviewer, as well as relevant works on legal knowledge representation and formal legal reasoning, which may be critical to developing more robust, transparent, and verifiable legal alignment.

---

> > > > ### Author Response · Authors · 2026-04-06
> > > > **Response (part 4)**
> > > >
> > > > ## 8. The role of institutional frameworks
> > > >
> > > > > The subsection on institutional frameworks is confusing. How do these frameworks differ from standard legal regulations that impose requirements on actors that produce AI?
> > > >
> > > > **The institutional frameworks we propose are distinct as they specifically aim to support legal alignment.** As we discuss in Section 2.2, there already exists a wide array of institutional frameworks for regulating AI technology, ranging from general law frameworks such as tort liability to AI-specific policy instruments such as the EU AI Act. The frameworks we discuss in Section 4.3 differ from these in that they focus specifically on supporting legal alignment, i.e., incentivizing or requiring that companies developing and deploying AI ensure their systems do in fact comply with law. For example, we propose establishing regulations that would require “[p]re-deployment legal alignment testing and post-deployment monitoring”. In practice, this framework could mandate that companies conduct evaluations to assess whether AI systems deployed in commercial applications engage in fraudulent misrepresentation or violate corporate governance laws. Critically, existing institutional frameworks (at least in the U.S. and EU) do not include such stipulations.
> > > >
> > > > **Action 8:** We have clarified the role of institutional frameworks in Section 4.3, including by using the above example.
> > > >
> > > > ---
> > > >
> > > > ## 9. The adaptability of law
> > > >
> > > > > The authors should discuss how a reactive-by-design framework like the law can adapt to a rapidly mutating AI landscape.
> > > >
> > > > **We recognize the challenges and opportunities arising from the (in)adaptability of law.** As noted by the reviewer, law’s “pacing problem” ([Collingridge, 1980](https://www.google.co.il/books/edition/The_Social_Control_of_Technology/hCSdAQAACAAJ?hl=en); [Marchant, 2011](https://link.springer.com/book/10.1007/978-94-007-1356-7)) is a central challenge for legal alignment. Section 3.2 attempts to explicitly grapple with this challenge by discussing how “[l]aws can be amended, repealed, or reinterpreted in response to changes in social, economic, or technological conditions”, engaging with seminal literature on the subject ([Holmes, 1897](https://doi.org/10.2307/1342108); [Lessig, 1993](https://chicagounbound.uchicago.edu/journal_articles/7788/); [Habermas, 1996](https://mitpress.mit.edu/9780262581622/between-facts-and-norms/)). In brief, our analysis, which draws on prior work by legal scholars, computer scientists, and philosophers ([O’Keefe et al., 2025](https://papers.ssrn.com/sol3/papers.cfm?abstract_id=5242643); [He et al., 2025](https://arxiv.org/abs/2509.01186); [Gabriel & Keeling, 2025](https://link.springer.com/article/10.1007/s11098-025-02300-4)) suggests that due to the diverse range of methods of legal change, legal alignment may prove highly adaptable to societal challenges arising from AI in the future. Nevertheless, in the final part of Section 5.3 we recognize that there will likely be some unavoidable constraints on the capacity of law to adapt to new and unforeseen scenarios arising from new developments in AI technology.
> > > >
> > > > **Action 9:** We have added in Section 3.2 further discussion of how law’s “pacing problem” could challenge the efficacy and desirability of legal alignment in the coming years.
> > > >
> > > > ---
> > > >
> > > > ## 10. Takeaway boxes
> > > >
> > > > > Adding quick "takeaway boxes" at the end of specific sections or subsections would help readers immediately grasp the core findings.
> > > >
> > > > **Action 10:** Thank you for suggesting this. We agree that “takeaway boxes” would be helpful and we have added them after Sections 2 through 5.
> > > >
> > > > ---
> > > >
> > > > ## 11. Broader impact statement
> > > >
> > > > > Adding a brief Broader Impact Statement clarifying that technical legal alignment does not absolve AI developers of their regulatory and legal liabilities would strengthen the work and mitigate this concern.
> > > >
> > > > **Action 11:** We appreciate this suggestion. We have added a Broader Impact Statement that includes these important clarifications.
> > > >
> > > > ---
> > > >
> > > > ## 12: Proofreading
> > > >
> > > > > The paper contains a few typos and grammatical errors (e.g., "sycophantic").
> > > >
> > > > **We have conducted another round of proofreading.** In addition, to clarify the specific issue raised, our use of the term “sycophantic” refers to AI systems that are flattering and people-pleasing, as used in [Sharma et al. (ICLR 2024)](https://openreview.net/forum?id=tvhaxkMKAn) and [Cheng et al. (Science 2026)](https://www.science.org/doi/10.1126/science.aec8352).
> > > >
> > > > ---
> > > >
> > > > ## Any additional feedback?
> > > >
> > > > **Thank you again for your detailed and constructive feedback. We believe and hope these changes address your core critiques of our paper. We would be happy to know what other changes we can make to further improve the paper, so please let us know if you have additional feedback. Thank you!**

---

### Review · Reviewer_Ek3z · 2026-04-27

**Summary Of Contributions:**

This paper introduces legal alignment as another approach towards AI alignment, to be potentially used in conjunction with other forms of AI alignment and legal regulation. Specifically, the authors define legal alignment, set it in the broader context of AI and the law and delineate the various motivations for this approach. They round out the paper with a survey of practical implementations of legal alignment and open questions that this approach raises.

**Additional Comments:**

While I appreciate the authors’ efforts in being nuanced in their discussion, there are several complications that are likely to be encountered with the legal alignment approach, primarily due to the variety of interpretations of laws, many of which morph over time and legal practitioners. Some of these questions include:

1.	On Page 2, the authors state that “legal rules are ideally a product of transparent and publicly accountable processes that are themselves generated by rules and procedures that a political community recognizes as legitimate”.

  a.	The operative word in this assertion is “ideally”. In practice, this may not hold true. How would any legal alignment approach be made robust to imperfections? If human overview is required, would that be viable when the rate of AI evaluations and decisions far outpace human capabilities?

  b.	How does legal alignment account for major adjustments to the law?

2.	On Page 3, it is stated that “law also contains relatively robust methods for balancing competing societal interests and adapting existing rules and principles to new economic and technological conditions”. This validity of this statement is in question when it comes to systemic biases, which have long been one of the challenges with AI models often known to amplify these biases.
3.	The question 1(a) also applies to the sentence on page 10 that reads “For example, legal decision-makers, particularly judges, construct meaning through various interpretive methodologies and the creation of precedent that can subsequently be used to resolve future cases”.
4.	On page 11, it is averred that “legal alignment could potentially reduce the prospect of Ai systems engaging in illegal conduct in the first place, provided the legal system targets the underlying conduct of concern.” This suggests that an AI model should be able to make legal assessments on the fly. This may not always be possible, particularly with issues that are open to legal interpretation. Wouldn’t AI-specific legal frameworks for such issues be required?

**Audience:**

Yes

**Audience Explanation:**

This have been addressed in the previous taxt box. In short, there is an inherent interest in guaranteeing that AI models behavior in a manner compliant with the law, especially when looking to deploy them in sensitive domains.

**Broader Impact Concerns:**

I think that the authors have adequately addressed the broader impacts of their work.

**Claims And Evidence:**

Yes

**Claims Explanation:**

The legal alignment paradigm seeks to “build” legality into AI models, so that these models are safer and law-abiding by nature. Predicated on the public legitimacy of the law and legal process, this alignment would serve to naturally yield models that are widely accepted by society. Additionally, legally aligned models would be deployable in sensitive domains, such as finance and healthcare.
T
hroughout the enclosed discussion, the authors have endeavored to emphasize the nuances of legal alignment and the complications that may arise due to the incorporation of legal principles into the reasoning of AI models. Allied with the assertion that legal alignment is not a substitute for existing AI alignment approaches, rather an additional means of alignment, there is clear value that it could be a potentially very valuable approach as AI proliferates to other domains.

I would also like to applaud the authors for the clarity of their writing, with clear sections for the introduction and broader context of legal alignment, motivations, implementation and open questions. The tables corresponding to each section also serves as an effective reference for a reader.

**Requested Changes:**

**Critical changes:**

On Page 13, the last paragraph of Section 3 that starts with “If, on the one hand, AI systems...”, the authors appear to be discussing a dichotomy in AI systems and the effect of legal alignment on them. However, these effects appear to be applicable to both types of AI systems. For instance, protection against potentially catastrophic harms and mitigation of ongoing harms arising from AI systems engaging in unlawful activity are consequences of legal alignment that should apply to AI systems, regardless of the rate of development.

**Suggested changes:**

In the discussion on Page 20 prefaced with the question “What interventions can support AI systems obeying both the letter and spirit of the law?”, there is an implicit assumption that AI models reason like humans, This may be outside the purview of this paper but in the event that AI reasoning does not alignment with humans, AI obeying the spirit of the law via legal alignment may not be possible.

---

> ### Author Response · Authors · 2026-05-04
> **Thanks + response (part 1)**
>
> We are thankful for your helpful comments and suggestions. We were glad to hear that you consider legal alignment of inherent interest and applauded the paper’s clarity of writing and nuanced discussion. **We have uploaded a further revised version of the paper with additional changes made in purple that we hope address your requests.**
>
> ---
>
> ## 1. Legal alignment across different AI systems and development timelines
>
> > the authors appear to be discussing a dichotomy in AI systems and the effect of legal alignment on them. However, these effects appear to be applicable to both types of AI systems … regardless of the rate of development.
>
> **We agree that legal alignment is beneficial across different types of AI systems and irrespective of the rate of AI development.** We appreciate the reviewer’s feedback that legal alignment can help mitigate risks arising from a diverse range of AI systems, and that its value is not contingent on the particular pace at which AI systems develop.
>
> **Action 1:** We have revised the last paragraph of Section 3.4 (p. 13) to reflect this important clarification.
>
> ---
>
> ## 2. Limitations of AI legal reasoning
>
> > in the event that AI reasoning does not align[] with humans, AI obeying the spirit of the law via legal alignment may not be possible.
>
> **We agree that (current) AI legal reasoning methods may not be sufficient for aligning AI systems with the “spirit” of the law.** Despite improvements in AI legal reasoning (e.g., [Zheng et al., 2025](https://dl.acm.org/doi/10.1145/3709025.3712219); [Han et al., 2025](https://aclanthology.org/2025.emnlp-main.1787/); [Posner & Saran, 2026](https://papers.ssrn.com/sol3/papers.cfm?abstract_id=6155012)), AI legal reasoning continues to face significant shortcomings, and differs markedly from human legal reasoning (e.g., [Doyle & Tucker, 2025](https://dl.acm.org/doi/10.1145/3709025.3712220); [Purushothama et al., 2025](https://aclanthology.org/2025.nllp-1.22/); [Pruss & Allen, 2025](https://ojs.aaai.org/index.php/AIES/article/view/36695)). As the reviewer pointed out, these shortcomings may pose obstacles to designing AI systems that follow the spirit of the law, and not merely the letter of the law. This is encapsulated in our discussion of how robust AI legal reasoning and capabilities are a prerequisite for legal alignment (see Section 2.2, p. 7).
>
> **Action 2:** We have revised the second paragraph of Section 5.2 (p. 22) to discuss how the limitations of current AI legal reasoning methods may be inadequate for designing AI systems that can understand and comply not only with the law’s formal expression, but also its underlying purpose.
>
> ---
>
> ## 3. “Imperfect” and biased law
>
> > “legal rules are ideally a product of transparent and publicly accountable processes … that a political community recognizes as legitimate”. … In practice, this may not hold true. How would any legal alignment approach be made robust to imperfections?
>
> > “law also contains relatively robust methods for balancing competing societal interests”. This … is in question when it comes to systemic biases, which have long been one of the challenges with AI models
>
> **We agree with these critical observations regarding the “imperfections” and biases of law, and address them extensively.** Specifically, in Section 5.1 (p. 21) we draw on seminal works in legal and political theory (e.g., [Cover, 1975](https://www.jstor.org/stable/j.ctt32bmbr); [Dyzenhaus, 2010](https://books.google.co.il/books?id=0w619eWQjNUC&source=gbs_navlinks_s)) that highlight how law can fail to fulfil its ideals of legitimacy, including through the enactment of unjust or oppressive laws, as well as laws that entrench or exacerbate existing social and economic inequalities.
>
> By way of further clarification, we note that:
> - Concerns regarding the legitimacy of law extend beyond the scope of our paper and are not unique to legal alignment, as we point out in Section 5.1.
> - Our paper does **not** suggest that legal alignment is a “perfect” or comprehensive solution to the potential societal harms from AI; rather, legal alignment aims to serve as a “lower bound” (p. 3), which should be complemented by additional alignment approaches that, for example, aim to support users’ long-term health and well-being (p. 21).
>
> **Action 3a:** We have added to the final paragraph of Section 5.1 (p. 21) additional references from the field of critical legal studies that characterize the systemic biases that pervade the legal system ([Bell, 1980](https://www.jstor.org/stable/1340546); [Unger, 1983](https://doi.org/10.2307/1341032); [Moyn, 2024](https://yalelawjournal.org/essay/reconstructing-critical-legal-studies)).
>
> **Action 3b:** Following the reviewer’s suggestion, we have explicitly clarified in Section 5.1 (p. 21) that systemic biases affect judicial making and the resulting case law it produces, and risk being amplified by AI systems.

---

> > ### Author Response · Authors · 2026-05-04
> > **Response (part 2)**
> >
> > ## 4. Changes in law
> >
> > > How does legal alignment account for major adjustments to the law?
> >
> > **The adaptability of law is essential to legal alignment and is one of its key advantages.** As pointed out by the reviewer, legal alignment will indeed have to contend with ongoing substantial changes to the content of law. We discuss this issue in the final paragraph of Section 3.2 (p. 11), covering the various mechanisms through which law evolves, including the enactment, repeal, and amendment of laws, as well as new interpretations of existing laws.
> >
> > On balance, following [O’Keefe et al. (2025)](https://papers.ssrn.com/sol3/papers.cfm?abstract_id=5242643), we consider the dynamic and evolving content of law a highly desirable property of legal alignment. As law changes in response to new social and technological conditions (e.g., the development of labor law in response to industrialization and environmental law following chemical-produced pollution), legal alignment will reflect and incorporate those changes, rather than remain beholden to outdated or inappropriate norms.
> >
> > **Action 4:** We have added further discussion in Section 3.2 (p. 11) regarding the technical challenges and methods involved in ensuring that legal alignment can incorporate major changes in law.
> >
> > ---
> >
> > ## 5. Automated legal alignment evaluation + AI-specific legal frameworks
> >
> > > If human overview is required, would that be viable when the rate of AI evaluations and decisions far outpace human capabilities?
> >
> > > an AI model should be able to make legal assessments on the fly. This may not always be possible … Wouldn’t AI-specific legal frameworks for such issues be required?
> >
> > **We agree that the speed of operation of AI systems requires developing new technical approaches to legal alignment.** In line with the reviewer’s observation, in the third paragraph of Section 3.1 (pp. 9-10) and the first paragraph of Section 5.2 (pp. 21-22) we discuss how manual (human) evaluations of legal alignment may fail where AI systems outpace human understanding and decision-making. In response, Section 3.1 proposes developing scalable oversight mechanisms that automate legal alignment evaluations. This, however, runs into the second issue raised by the reviewer: namely, that AI models may lack the technical ability to conduct such evaluations in real-time. We share this concern, particularly in light of studies that illustrate the shortcomings of contemporary AI models in evaluating legal compliance (e.g., [Guldimann et al., 2024](https://arxiv.org/abs/2410.07959); [Marino et al., 2025](https://arxiv.org/abs/2510.01474)).
> >
> > In addition, **we agree that AI-specific legal frameworks may be needed, particularly to facilitate legal alignment testing** and, in turn, incentivize the development of AI systems that can reason about legal compliance with sufficient precision and speed. Section 4.3 provides an outline of the characteristics of such legal frameworks, including institutional mechanisms for facilitating technical evaluations, oversight, and enforcement. (For the avoidance of doubt, as per our clarification on p. 3, we do **not** consider legal alignment a substitute for legal regulation).
> >
> > **Action 5a:** We have revised Section 4.2 (p. 17) to explicitly refer to “Speed” as a criterion for the technical interventions needed to implement legal alignment.
> >
> > **Action 5b:** We have revised Section 4.3 (p. 18) to discuss how legal frameworks can incentivize the development of the technical abilities that are a prerequisite for developing legally aligned AI systems.
> >
> > ---
> >
> > ## Any additional feedback?
> >
> > **Thank you again for your detailed and constructive feedback. We believe and hope these changes address your requests. We would be happy to know what other changes we can make to further improve the paper, so please let us know if you have additional feedback. Thank you!**

---

### Decision · Action_Editor_i7ep · 2026-05-17

**Recommendation:** Accept as is

**Audience:**

Yes

**Audience Explanation:**

This paper should be of interest to at least some members of the TMLR audience. Legal alignment lies at the intersection of AI alignment, AI safety, AI governance, and legal-institutional research. As AI systems are increasingly deployed in finance, healthcare, law, governance, and other high-risk domains, it is practically important to consider how AI systems can not only conform to abstract human preferences, but also comply with legal norms at the level of system design and operation.

Although some parts of the paper are strongly legal in nature, the authors have improved its readability and relevance for AI researchers and practitioners by adding a more explicit technical research agenda. Therefore, TMLR readers interested in AI safety, AI governance, and sociotechnical ML are likely to find valuable research perspectives and future directions in this paper.

**Claims And Evidence:**

Yes

**Claims Explanation:**

The main claims of this paper are supported by reasonably sufficient, clear, and credible evidence. The paper substantiates its core arguments through interdisciplinary literature synthesis, legal-theoretical analysis, a review of research on AI alignment and AI governance, and structured conceptual reasoning. It clearly explains the object of study of legal alignment, its three core pathways, and the empirical evaluations, technical interventions, and institutional frameworks needed for its implementation. These main claims are further supported by extensive literature from law, AI safety, AI alignment, and AI governance.

Although the initial submission had some issues, such as insufficiently concrete discussion of technical implementation, an unclear scope regarding AI versus LLMs/GenAI, and a lack of worked examples and case studies, the authors have made substantial revisions during the rebuttal process. Therefore, I believe that the paper’s support for its main claims meets TMLR’s expectations for a survey-style submission.